# Instance-wise Adaptive Scheduling via Derivative-Free Meta-Learning

**Hefang Qing**[1†], **Miao Zhang**[2†], **Yaoxin Wu**[3], **Weinan Huang**[2],
**Jianhao Yang**[2*], **Wen Song**[1*], **Gang Wang**[4]
[1]Shenzhen Research Institute of Shandong University    [2]Xiaomi Corporation
[3]Eindhoven University of Technology   [4]Zhongguancun Academy
`qinghefang@email.sdu.edu.cn  zhangmiao20@xiaomi.com`
`y.wu2@tue.nl  huangweinan3@xiaomi.com  yangjianhao@xiaomi.com`
`wensong@email.sdu.edu.cn  gangwang@umn.edu`

## Abstract

Deep reinforcement learning has achieved remarkable progress in solving NP-hard scheduling problems. However, existing methods primarily focus on optimizing average performance over training instances, overlooking the core objective of solving each individual instance with high quality. While several instance-wise adaptation mechanisms have been proposed, they are test-time approaches only and cannot share knowledge across different adaptation tasks. Moreover, they largely rely on gradient-based optimization, which could be ineffective in dealing with combinatorial optimization problems. We address the above issues by proposing an instance-wise meta-learning framework. It trains a meta model to acquire a generalizable initialization that effectively guides per-instance adaptation during inference, and overcomes the limitations of gradient-based methods by leveraging a derivative-free optimization scheme that is fully GPU parallelizable. Experimental results on representative scheduling problems demonstrate that our method consistently outperforms existing learning-based scheduling methods and instance-wise adaptation mechanisms under various task sizes and distributions.

## 1 Introduction

Scheduling aims to optimize resource allocation for task completion within specified time constraints, playing a pivotal role in a wide range of practical domains such as manufacturing, logistics, and healthcare (Khadivi et al., 2025). As two fundamental models, Job-shop Scheduling Problem (JSP) and its extension, Flexible Job-shop Scheduling Problem (FJSP) receive much attention. However, solving JSP and FJSP optimally remains a significant challenge due to their well-known NP-hardness (Michael, 1995; Mazyavkina et al., 2021). Especially for industrial level large-scale instances, exact algorithms such as Mixed-Integer Programming (MIP) and Constraint Programming (CP) are often prohibitive due to the excessive computational cost (Da Col & Teppan, 2019). Heuristic and metaheuristic methods could strike a balance between solution quality and computational time, but they are typically less accurate due to their reliance on predefined rules and lack of adaptability to specific scenarios (Li et al., 2024a).

Recently, Deep Reinforcement Learning (DRL), as an emerging alternative method (Feng et al., 2026), has been successfully applied to complex scheduling problems. A notable direction is to use DRL to learn Priority Dispatching Rules (PDRs) (Zhang et al., 2020; Song et al., 2023; Wang et al., 2023). The learned policies are often superior to manually designed PDRs, and can generate solutions within short run time (Mazyavkina et al., 2021). However, the solution quality of existing learning based scheduling methods are still relatively far from optimaliy. One reason is that they focus on training a deep policy model to optimize its average performance over training instances. Given that currently DRL can only obtain suboptimal policy, this means a well-trained policy can still produce

---

[†]These authors contributed equally to this work.
[*]Corresponding authors.

poor solutions for some testing instances, even when they come from the training distribution (Wang & Li, 2023).

One way to overcome this limitation is test-time adaptation, which is to fine-tune the pre-trained model on each specific instance being solved. Active Search (AS) (Bello et al., 2016) and Efficient Active Search (EAS) (Hottung et al., 2021b) are two representative methods, and the latter shows good performance on JSP. However, this popular paradigm suffers from two limitations. First, it works as a pure test-time mechanism that fine-tunes the model on each instance separately, which is inefficient since the adaptation knowledge is discarded and not reusable. Second, existing works use gradient-based methods for adaptation, which performs well in conventional deep (reinforcement) learning tasks but could fall short when dealing with the instance-wise search task for complex combinatorial optimization problems such as JSP and FJSP, since they could easily fall into local optimum.

In this paper, we address the above issues by proposing a meta-learning method driven by derivative-free optimization, for solving complex scheduling problems. Our first contribution is an instance-wise meta-learning framework based on Model-Agnostic Meta-Learning (MAML) (Finn et al., 2017), which simulates the fine-tuning process during training so as to obtain a meta-model that explicitly considers the needs of fine-tuning, thus providing a well-initialized model for each new instance. This framework is model-agnostic, and is generally applicable to a wide range of deep policy model. Next, motivated by the recent success of evolutionary strategies in DRL (Salimans et al., 2017; Song et al., 2020; Kirsch et al., 2022), we design a fully derivative-free method to train the meta-model, which not only overcomes the limitation of gradient-based methods in instance-wise searching but also bypasses the complicated gradient computation in the original MAML. We design two Monte Carlo (MC) strategies for gradient estimation in the inner loop, which effectively improves the training performance. Finally, we design a population-parallel framework that shifts the CPU-intensive computational tasks in traditional evolutionary strategies (Salimans et al., 2017) to GPU parallel processing, significantly reducing the training overhead.

We validate the effectiveness of our method mainly on FJSP, which is much harder than JSP. Specifically, we deploy our method to state-of-the-art FJSP PDR learning model in (Wang et al., 2023). Experimental results demonstrate that the instance-wise fine-tuning strategy significantly improves the model's adaptability to unseen test instances. Moreover, our approach consistently achieves superior performance across benchmark datasets of varying sizes and distributions, outperforming existing instance-level adaptation methods. We also extend our method to JSP in a non-reinforcement learning setting (Corsini et al., 2024), showcasing its strong compatibility and adaptability across different learning paradigms.

## 2 RELATED WORK

**Learning based scheduling.** Motivated by the recent success of Neural Combinatorial Optimization (NCO) (Bengio et al., 2021), researchers have begun to utilize deep (reinforcement) learning to tackle JSP and FJSP. The most popular paradigm is PDR learning, which formulates the PDR based schedule construction as a Markov Decision Process (MDP), and uses DRL to automatically train the scheduling policy. A common choice in this direction is to represent construction states based on disjunctive graph (e.g., (Zhang et al., 2020; Park et al., 2021; Song et al., 2023; Teichteil-Königsbuch et al., 2023)), and design Graph Neural Network (GNN) based policy network to achieve size-invariance. Besides GNN, other types of neural architecture based on Pointer Network (Corsini et al., 2024) and Attention Mechanism (Wang et al., 2023; Chen et al., 2022; Pirnay & Grimm, 2024a;b) have also been proposed and can achieve even better performance. Another direction is to learn control policies for local search algorithms (Zhang et al., 2024a;b), which tend to deliver better solutions than PDR learning methods at the cost of longer run time.

**Instance-wise adaptation.** Above methods follow the convention in machine learning to optimize the average performance over training instances, which often leads to suboptimal performance on unseen instances. This could be alleviated by Active Search (AS) (Bello et al., 2016) which dynamically adjusts pre-trained model parameters on each testing instance. Efficient Active Search (EAS) (Hottung et al., 2021b) improves AS by updating only a subset of model parameters to reduce computational costs. However, AS and EAS do not change the original training objective of average performance and the adaptation is purely test-time. Meta-models for more efficient adaptation have been explored

in other combinatorial optimization problems such as vehicle routing and graph optimization (Qiu et al., 2022; Wang & Li, 2023; Son et al., 2023), but they rely on problem-specific techniques and are not directly applicable here. Moreover, these methods use gradient-based optimization for fine-tuning, whereas our method avoids strong assumptions about the problem or neural architecture and is the first to employ gradient-free optimization for such tasks. Additionally, some research performs instance-wise search in a learned continuous space for high-quality solution distributions (Hottung et al., 2021a; Li et al., 2023; 2024b), but these methods focus less on adaptation, and models are not updated per instance. Our approach is orthogonal to these works and could potentially be combined for better performance.

## 3 PRELIMINARIES

**Job-shop scheduling problem** involves a set of jobs $J = \{J_1, J_2, \ldots, J_n\}$ to be processed on machines $M = \{M_1, M_2, \ldots, M_m\}$, where each job $J_i$ consists of a sequence of operations $O_i = \{O_{i1}, O_{i2}, \ldots, O_{in_i}\}$ to be assigned to machines. In JSP, each operation can only be assigned to one specific machine, whereas FJSP allows multiple machine options. The goal is to minimize the makespan, defined as the maximum completion time across all operations, i.e., $C_{\max} = \max_{i,j}\{C_{ij}\}$.

In this paper, we use the PDR learning model for FJSP in (Wang et al., 2023) to evaluate the problem. The model treats scheduling as a sequential decision task. At each step $t$, a neural network receives the state $s_t$ (including operations and machines) and outputs an action $a_t$ that assigns an unscheduled operation to an available machine. This repeats until all operations are scheduled, with rewards based on the final makespan. The policy $\pi_\theta(a_t|s_t)$ is trained using Proximal Policy Optimization (PPO) (Schulman et al., 2017) to maximize cumulative reward. After training, the policy can be used in greedy or sampling modes. Greedy selects the best action at each state, while sampling generates multiple schedules and returns the best solution, though at a higher computational cost.

**Derivative-free optimization (DFO)**, also known as zero-order optimization, encompasses methods that do not rely on gradients. DFO is effective in optimization problems where gradients are costly or inaccessible. Here we use OpenAI's Evolutionary Strategy (ES) (Salimans et al., 2017) for optimization, which belongs to the class of Natural Evolution Strategies (NES) (Wierstra et al., 2014). Let $F$ be the objective function over parameter vector $\theta$. NES models the population as a distribution $p_\psi(\theta)$, parameterized by $\psi$. The optimization process aims to maximize the expected objective $\mathbb{E}_{\theta \sim p_\psi}[F(\theta)]$ by updating $\psi$ via stochastic

---

**Algorithm 1** ES Gradient Estimation

---

**Require:** Neural network parameter set $\theta$, noise standard deviation $\sigma$, population size $\mu$;
1: Sample : $\varepsilon_1, \ldots, \varepsilon_\mu \sim \mathcal{N}(0, I)$
2: **for** $i = 1, \ldots, \mu$ **do**
3:     Compute fitness: $F_i = F(\theta + \sigma\varepsilon_i)$
4: **end for**
5: **return** $\frac{1}{\mu\sigma} \sum_{i=1}^{\mu} F_i \varepsilon_i$

---

gradient ascent, with the gradient $\nabla_\psi \mathbb{E}_{\theta \sim p_\psi}[F(\theta)] = \mathbb{E}_{\theta \sim p_\psi}[F(\theta)\nabla_\psi \log p_\psi(\theta)]$. To avoid non-smoothness from the environment or discrete policy actions, we follow OpenAI's implementation to use an isotropic multivariate Gaussian distribution for the population, with mean $\psi$ and fixed covariance $\sigma^2 \mathbf{I}$ (Salimans et al., 2017). This allows the expected objective to be expressed as $\mathbb{E}_{\theta \sim p_\psi}[F(\theta)] = \mathbb{E}_{\varepsilon \sim \mathcal{N}(\mathbf{0}, \mathbf{I})}[F(\theta + \sigma\varepsilon)]$. We optimize over $\theta$ directly using stochastic gradient ascent: $\nabla_\theta \mathbb{E}_{\varepsilon \sim \mathcal{N}(\mathbf{0}, \mathbf{I})}[F(\theta + \sigma\varepsilon)] = \frac{1}{\sigma} \mathbb{E}_{\varepsilon \sim \mathcal{N}(\mathbf{0}, \mathbf{I})}[F(\theta + \sigma\varepsilon)\varepsilon]$, which can be approximated through the Monte Carlo sampling procedure in Algorithm 1.

## 4 METHODOLOGY

In this section, we propose a general instance-wise meta-learning framework for well-adapted parameter initialization in downstream fine-tuning, fully exploiting the potential of derivative-free optimization in instance-level adaptation tasks. Additionally, we design an efficient population-based parallelization strategy that significantly enhances computational efficiency.

### 4.1 INSTANCE-WISE DERIVATIVE-FREE META-LEARNING FRAMEWORK

We define a FJSP instance class as $\Omega = (n, m, o, t)$, where $n$ and $m$ denotes the number of jobs and machines, $o = [o_{\min}, o_{\max}]$ specifies the range of the number of operations in each job, while

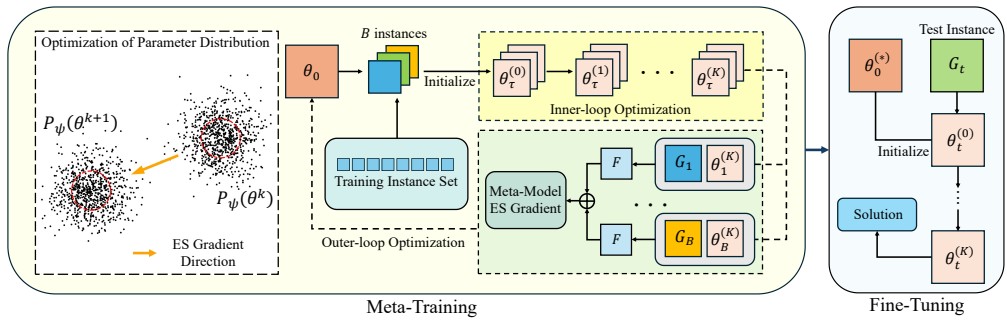

Figure 1: overall framework of our method

$t = [t_{\min}, t_{\max}]$ defines the processing time range for operations executed on different machines. We denote $G_\tau \in \Omega$ as a specific scheduling instance sampled from the defined instance class.

Unlike traditional methods that aim to learn a single model with optimal average performance over training set, our goal is to learn an initialization of model parameters that enables efficient adaptation to unseen instances at test time. This initialization allows the model to rapidly converge to high-quality, instance-specific solutions through fine-tuning during inference, which offers more potential than test-time only methods such as AS (Bello et al., 2016) and EAS (Hottung et al., 2021b). Accordingly, our training problem is:

$$\theta_0^* = \arg\min_{\theta_0} \mathbb{E}_{G_\tau \sim \Omega} \left[ F \left( \theta_\tau^{(K)} \,\middle|\, G_\tau \right) \right] \tag{1}$$

where $\theta_0$ denotes the *meta-model* that provides a starting parameter for each instance, and $\theta_\tau^{(K)}$ is the fine-tuned model obtained after $K$ steps of gradient updates from $\theta_0$ on instance $G_\tau$, which is then used to generate the final solution (i.e., schedule) for $G_\tau$.

We implement this learning task using the MAML framework (Finn et al., 2017). The meta-training process consists of two stages: inner-loop optimization and outer-loop optimization. The inner loop performs simulated fine-tuning on each individual instance to capture instance-specific features and enable rapid adaptation. The outer loop aggregates feedback from multiple instances to update the meta-model parameters, thereby enhancing its ability to quickly adapt to new, unseen instances. The pseudocode for the meta-training process is presented in Algorithm 2.

---

**Algorithm 2** Instance-wise Derivative-Free Meta-Training

---

**Require:** Training instance set $\Omega$, mini-batch size $B$, number of inner-loop updates $K$, adaptation step size $\alpha$, meta step size $\beta$, noise standard deviation $\sigma$, number of epochs $T$;
1: Randomly initialize the meta-model $\theta_0$
2: **for** $t = 1, \ldots, T$ **do**
3:     **for** each randomly sampled instance $G_\tau \in \Omega$, $\tau = 1, \ldots, B$ **do**
4:         Initialize instance-specific model: $\theta_\tau^{(0)} \leftarrow \theta_0$
5:         **for** $k = 1, \ldots, K$ **do**
6:             $\theta_\tau^{(k)} \leftarrow \theta_\tau^{(k-1)} - \alpha \nabla_{\theta_\tau^{(k-1)}} \mathbb{E}_{\varepsilon \sim \mathcal{N}(\mathbf{0}, \mathbf{I})} \left[ F \left( \theta_\tau^{(k-1)} + \sigma\varepsilon \,\middle|\, G_\tau \right) \right]$
7:         **end for**
8:     **end for**
9:     $\theta_0 \leftarrow \theta_0 - \frac{\beta}{B} \sum_{\tau=1}^{B} \nabla_{\theta_0} \mathbb{E}_{\varepsilon \sim \mathcal{N}(\mathbf{0}, \mathbf{I})} \left[ F \left( \theta_\tau^{(K)} + \sigma\varepsilon \,\middle|\, G_\tau \right) \right]$
10: **end for**

---

A key issue in Algorithm 2 is how to compute the inner-loop and outer-loop gradients in Line 6 (computing gradients of the instance-wise model) and Line 9 (computing gradients of the meta-model defined in Eq. 1). Unlike the mainstream gradient-based methods (e.g. (Manchanda et al., 2022; Zhou et al., 2023; Qiu et al., 2022; Wang & Li, 2023)), we propose a full derivative-free method to estimate these gradients. Leveraging the black-box nature of DFO, our framework relies solely on the evaluability of the objective function, and makes no assumptions about the underlying MDP and the

environment's structural priors, therefore is easy to implement and potentially applicable to various scheduling problems. Moreover, using DFO in the inner loop is beneficial for finding high-quality solution for each instance due to its strong global search ability. The overall architecture is illustrated in Figure 1.

### 4.1.1 INNER-LOOP OPTIMIZATION

In our approach, inner-loop optimization updates instance-specific parameters. For a training instance $G_\tau$, its model is initialized to the meta-model, i.e., $\theta_\tau^{(0)} \leftarrow \theta_0$. Then, $K$ steps of ES gradient updates are applied to obtain a model $\theta_\tau^{(K)}$. At each step $k$, a population of $\mu$ individuals is sampled from the parameter distribution $p_\psi(\theta_\tau^{(k)})$, and their fitness $F_i$ is evaluated. The ES gradient is computed according to Algorithm 1 and used to update the instance-specific model $\theta_\tau$.

In standard NES (Wierstra et al., 2014; Salimans et al., 2017), fitness $F_i$ is computed by a single policy rollout. However, this single-sample strategy is unstable in complex scheduling problems due to environmental stochasticity. To address this, we propose a parallel sampling mechanism, generating $L$ solutions per individual $i$ by sampling its policy network $\theta + \sigma\varepsilon_i$, for more accurate gradient estimation.

**MC averaging estimation.** Our first method computes the mean of the $L$ objective values to obtain the individual's fitness, meaning that we use the following equation for fitness computation in Line 3 in Algorithm 1:

$$F_i = F(\theta + \sigma\varepsilon_i) = \frac{1}{L} \sum_{l=1}^{L} F_i^{(l)}(\theta + \sigma\varepsilon_i) \tag{2}$$

where $F_i^{(l)}(\theta + \sigma\varepsilon_i)$ is the objective value (i.e., makespan) of the $l$-th solution. This simple strategy effectively reduces variance in gradient estimation and improves training performance. However, when the policy is used in the sampling mode, MC averaging is less effective since the mean value cannot reflect the sampling result. To further enhance performance under sampling, we design another ES gradient estimator that incorporates sampling information into the inner-loop optimization process.

**MC best-sample estimation.** To better align with the sampling mode commonly used for NCO policies, we replace the averaging of objective values across multiple feasible scheduling solutions with a best-sample-based strategy. Specifically, during each inner-loop update, we exclusively utilize the best objective value among the $L$ solutions of each individual as its fitness estimate for ES gradient computation. The fitness $F_i$ of each individual in Algorithm 1 is then computed as follows:

$$F_i = F(\theta + \sigma\varepsilon_i) = \min_{l=1,\ldots,L} \left\{ F_i^{(l)}(\theta + \sigma\varepsilon_i) \right\}. \tag{3}$$

### 4.1.2 OUTER-LOOP OPTIMIZATION

In Algorithm 2, the aim of the outer-loop optimization is to update the meta-parameters that govern the model's ability to rapidly adapt. Unlike conventional task-distribution-based meta-learning approaches that primarily focus on task-level generalization, our method shifts the optimization emphasis toward improving the model's adaptability to individual instances. The outer loop aggregates adaptation outcomes from the inner loop across multiple training instances to optimize the shared meta-parameters, aiming to maximize the expected performance of the model after instance-level fine-tuning. Specifically, for each training instance (treated as a pseudo-test instance) we optimize the meta-model by

---

**Algorithm 3** Meta-Level ES Gradient Estimation

**Require:** Population size $\mu$, adaptation steps $K$, adaptation step size $\alpha$, noise standard deviations $\sigma, \eta$;
1: Sample $\varepsilon_1, \ldots, \varepsilon_\mu \sim \mathcal{N}(0, I)$, $g_1, \ldots, g_\mu \sim \mathcal{N}(0, I)$
2: **for** $j = 1, \ldots, \mu$ **do**
3:     **for** $k = 1, \ldots, K$ **do**
4:         **for** $i = 1, \ldots, \mu$ **do**
5:             $F_{(i,j)} = F(\theta_{k-1} + \eta g_j + \sigma\varepsilon_i)$
6:         **end for**
7:         $\theta_k + \eta g_j \leftarrow \theta_{k-1} + \eta g_j - \alpha \cdot \frac{1}{\mu\sigma} \sum_{i=1}^{\mu} F_{(i,j)}\varepsilon_i$
8:     **end for**
9:     $F_j = F(\theta_K + \eta g_j)$
10: **end for**
11: **return** $\frac{1}{\mu\eta} \sum_{j=1}^{\mu} F_j \cdot g_j$

---

maximizing the performance after $K$ steps of ES gradient-based adaptation. The computation of the meta ES gradient on a single instance is detailed in Algorithm 3. During training, the meta-parameters are updated once after simulating the $K$-step adaptation process on a mini-batch of $B$ instances $\{G_\tau\}_{\tau=1}^{B}$, as in Line 9 of Algorithm 2.

**First-order Approximations.** In our framework, DFO removes the need for second-order derivatives, but both optimization loops remain population-based, introducing perturbations to form a distribution of candidate solutions. This "perturbation-on-perturbation" mechanism resembles higher-order differentiation, increasing the variance of ES gradient estimates. Moreover, given that derivative-free methods are intrinsically slower to converge, the resulting computational overhead renders the training process practically infeasible in real-world applications.

We address this challenge by utilizing the first-order approximation scheme in FOMAML (Finn et al., 2017), which explicitly discards second-order terms and updates the meta-parameters using only first-order gradients. Although originally developed for gradient-based methods, it can be analogously applied to our derivative-free setting. Specifically, the first-order approximation of meta-model update can be expressed as:

$$\theta_0 \leftarrow \theta_0 - \frac{\beta}{B}\sum_{\tau=1}^{B}\nabla_{\theta_\tau}\mathbb{E}_{\varepsilon\sim\mathcal{N}(\mathbf{0},\mathbf{I})}\left[F\left(\theta_\tau^{(K)} + \sigma\varepsilon \middle| G_\tau\right)\right]. \tag{4}$$

We use the above equation to replace the meta-model update in Line 9 of Algorithm 2. The only notation difference is that we replace the full meta-gradient $\nabla_{\theta_0}$ in Line 9 of Algorithm 2 with the instance-specific gradient $\nabla_{\theta_\tau}$, which is the essence of first-order approximation and significantly reduces the computational overhead.

## 4.2 GPU-BASED PARALLELIZATION

Population-based DFO methods such as ES are naturally parallelizable. However, traditional implementations (Salimans et al., 2017; Song et al., 2020) typically rely on CPU clusters and distributed schedulers such as Dask or Ray for fitness evaluation and population evolution, limiting their ability to fully utilize the parallization power of GPU. The associated communication and synchronization overheads also become major performance bottlenecks.

In this paper, we develop a GPU-based framework for population-level parallel fitness evaluation, as illustrated in Figure 2, which significantly improves the computational efficiency of ES by parallelizing Line 2-4 in Algorithm 1 on GPU. The core idea is to replace the conventional per-individual evaluation scheme with a batch inference mechanism applied to the entire population. We adapt the neural network architecture to process all individuals' input states in a single forward pass, thereby achieving model-level full parallelism. Spefically, we construct a perturbation matrix $\boldsymbol{\varepsilon} = [\varepsilon_1, \ldots, \varepsilon_\mu] \in \mathbb{R}^{d\times\mu}$, where each column $\varepsilon_i \sim \mathcal{N}(0, \mathbf{I})$ is the Gaussian noise added to the mean parameter vector $\theta$ (of dimension $d$) to generate the parameter vector of the $i$-th individual, $\theta_i$. We then construct a population network, represented in matrix form as:

$$\Theta = \theta + \sigma\boldsymbol{\varepsilon} \tag{5}$$

The population network performs parallel forward propagation to produce action policies for all individuals, denoted as $\Pi = \mathcal{F}(\Theta, S)$, where $S = [s_1, \ldots, s_\mu]$ is the collection of input states for each individual, and $\Pi$ is the resulting policy output matrix. The network is used solely for efficient inference and does not involve backpropagation.

Furthermore, we vectorize the FJSP environment to support population-level parallel interactions. We construct a vector of environments $\mathcal{E} = [E_1, ..., E_\mu]$, which receives the full batch of policy outputs $\Pi$ and synchronously executes the interaction process for each individual in the environment and collect the corresponding individual fitness as $\mathcal{R} = \mathcal{E}(\Pi)$, where $\mathcal{R} = [r_1, ..., r_\mu] \in \mathbb{R}^\mu$ denotes the fitness vector of the population. Then we can compute the ES gradient in Algorithm 1 as:

$$\nabla_\theta \mathbb{E}_{\varepsilon\sim\mathcal{N}(\mathbf{0},\mathbf{I})}[F(\theta + \sigma\varepsilon)] = \frac{1}{\mu\sigma}\mathcal{R}\boldsymbol{\varepsilon}^\top \tag{6}$$

which is subsequently used to update the model parameters.

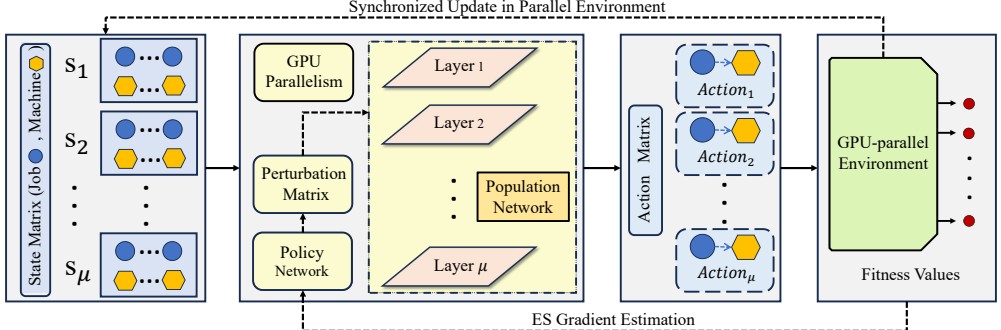

Figure 2: Overview of the GPU-based parallel ES framework

With this mechanism, the entire fitness evaluation process is offloaded to GPU, substantially reducing CPU dependence and data transfer overhead, while improving overall efficiency and system through-put. We will demonstrate the advantage of this implementation in the experiments. Remarkably, our approach even outperforms the speed-focused EAS algorithm on most instances.

## 5 EXPERIMENTAL RESULTS

In this section, we perform evaluation on FJSP which is harder than JSP. We apply our method to the state-of-the-art FJSP PDR learning model, Dual-Attention Network based Reinforcement Learning (DANIEL) (Wang et al., 2023). In Appendix D, we also provide an evaluation using the self-supervised training JSP model, Self-labeling Pointer Network (SPN) (Corsini et al., 2024), to demonstrate the versatility of our method. Our code is available at: `https://github.com/calmQ/DF-META`.

### 5.1 EXPERIMENTAL SETUP

**Dataset.** Following (Wang et al., 2023), we generate six group of synthetic FJSP instance of sizes $10 \times 5$, $15 \times 10$, $20 \times 5$, $20 \times 10$, $30 \times 10$, and $40 \times 10$ for training and evaluation. Two types of datasets are generated: SD1 (following (Song et al., 2023)) allows each job to contain a variable number of operations, increasing structural diversity; while SD2 (adopted from (Wang et al., 2023)) fixes the number of operations per job but significantly broadens the processing time range across alternative machines for each operation, thereby increasing scheduling complexity. Model training is conducted on the four smaller sizes, while the two larger sizes are reserved to assess generalization capability. For testing, besides synthetic datasets, we also use four public benchmarks that differ substantially in size (ranging from $10 \times 5$ to $30 \times 10$) and distribution, to assess cross-distribution generalization ability, including the ten classic mk instances (mk01 to mk10) from (Brandimarte, 1993), and three groups of la instances from (Hurink et al., 1994), namely rdata, edata, and vdata, each containing 40 instances. More details can be found in Appendix A.

**Training setup.** For our method, we set the population size $\mu = 100$, noise standard deviation $\sigma = 0.2$. Step size of the inner and outer loops are set to $\alpha = \beta = 5 \times 10^{-2}$. Each epoch generates $B = 20$ training instances, and the inner loop is executed for $K = 3$ steps per instance. The training runs for a total of 200 epochs. During the inner-loop adaptation phase, for the two proposed gradient estimation methods, the number of parallel samples per population member is set to $L = 20$ for the MC averaging and $L = 100$ for the MC best-sample. All hyperparameters are tuned on the smallest instance size ($10 \times 5$) and kept fixed across all instance sizes. All experiments are conducted on a workstation with an Intel Core i9-9900K CPU and a single NVIDIA RTX 4090 GPU.

**Baselines.** We use four types of baselines for comparison: 1) Google OR-Tools (Da Col & Teppan, 2019), a high-performing exact constraint optimization solver with 3600 seconds run time limit; 2) The best-performing manual PDR Most Work Remaining (MWKR) (Brandimarte, 1993; Montazeri & Van Wassenhove, 1990) as reported in (Song et al., 2023; Wang et al., 2023); 3) the original DANIEL model (Wang et al., 2023); and 4) two test-time fine-tuning strategies, Active Search (AS) (Bello et al., 2016) and Efficient Active Search (EAS) (Hottung et al., 2021b). For the latter,

Table 1: Performance evaluation on synthetic test sets. Sizes marked with $^*$ were unseen in training.

| | | $10 \times 5$ Obj. (Gap) | Time | $20 \times 5$ Obj. (Gap) | Time | $15 \times 10$ Obj. (Gap) | Time | $20 \times 10$ Obj. (Gap) | Time | $30 \times 10^*$ Obj. (Gap) | Time | $40 \times 10^*$ Obj. (Gap) | Time |
|---|---|---|---|---|---|---|---|---|---|---|---|---|---|
| SD1 | Ortools | 96.3 (0.00%) | 0.91h | 188.3 (0.00%) | 1h | 146.0 (0.00%) | 0.95h | 196.2 (0.00%) | 1h | 275.8 (0.00%) | 1h | 367.2 (0.00%) | 1h |
| | MWKR | 113.2 (17.55%) | 0.16s | 209.7 (11.36%) | 0.32s | 171.1 (17.19%) | 0.50s | 216.1 (10.14%) | 0.71s | 312.9 (13.45%) | 1.09s | 414.9 (12.99%) | 1.50s |
| Greedy | DANIEL | 106.7 (10.80%) | 0.45s | 197.6 (4.94%) | 0.94s | 161.3 (10.48%) | 1.35s | 198.5 (1.17%) | 1.85s | 281.5 (2.07%) | 2.76s | 371.5 (1.17%) | 3.77s |
| | AS | 104.9 (8.93%) | 0.27m | 193.9 (2.97%) | 0.47m | 156.9 (7.47%) | 0.85m | 194.7 (-0.76%) | 1.19m | 278.4 (0.94%) | 2.17m | 368.6 (0.38%) | 2.65m |
| | EAS | 103.7 (7.68%) | 0.22m | 194.0 (3.03%) | 0.42m | 156.4 (7.12%) | 0.74m | 194.5 (-0.87%) | 1.10m | 278.5 (0.98%) | 1.78m | 368.4 (0.33%) | 2.55m |
| | Ours | **103.1 (7.06%)** | 0.14m | **190.2 (1.01%)** | 0.31m | **153.6 (5.21%)** | 0.55m | **192.1 (-2.09%)** | 0.79m | **275.3 (-0.18%)** | 1.62m | **364.6 (-0.71%)** | 2.55m |
| Sampling | DANIEL | 101.7 (5.61%) | 0.74s | 192.8 (2.39%) | 1.87s | 153.2 (4.93%) | 3.89s | 193.9 (-1.17%) | 6.35s | 279.2 (1.23%) | 12.37s | 370.5 (0.90%) | 21.38s |
| | AS | 100.5 (4.36%) | 0.31m | 191.5 (1.70%) | 0.63m | 151.5 (3.77%) | 1.21m | 192.5 (-1.89%) | 1.80m | 277.9 (0.76%) | 3.42m | 368.5 (0.35%) | 5.12m |
| | EAS | 100.3 (4.15%) | 0.26m | 191.9 (1.91%) | 0.55m | 151.4 (3.70%) | 1.10m | 192.5 (-1.89%) | 1.61m | 277.7 (0.69%) | 2.95m | 368.5 (0.35%) | 4.71m |
| | Ours | **99.5 (3.32%)** | 0.19m | **188.6 (0.16%)** | 0.41m | **149.0 (2.05%)** | 0.82m | **189.1 (-3.62%)** | 1.34m | **273.4 (-0.87%)** | 2.65m | **363.0 (-1.14%)** | 4.55m |
| SD2 | Ortools | 326.2 (0.00%) | 0.51h | 597.7 (0.00%) | 1h | 376.9 (0.00%) | 0.77h | 461.9 (0.00%) | 1h | 669.2 (0.00%) | 1h | 938.3 (0.00%) | 1h |
| | MWKR | 549.4 (68.45%) | 0.16s | 1026.3 (71.76%) | 0.33s | 830.1 (120.27%) | 0.52s | 1041.1 (125.44%) | 0.71s | 1540.6 (130.22%) | 1.09s | 2036.5 (117.04%) | 1.50s |
| Greedy | DANIEL | 408.4 (25.20%) | 0.44s | 671.0 (12.27%) | 0.90s | 591.2 (56.86%) | 1.36s | 610.1 (32.09%) | 1.79s | 774.6 (15.75%) | 2.75s | 962.6 (2.59%) | 3.74s |
| | AS | 392.0 (20.17%) | 0.25m | 644.7 (7.87%) | 0.47m | 557.7 (47.97%) | 0.86m | 571.5 (23.73%) | 1.22m | 737.6 (10.22%) | 2.17m | 927.6 (-1.14%) | 2.62m |
| | EAS | 380.5 (16.65%) | 0.22m | 640.8 (7.22%) | 0.43m | 547.7 (45.32%) | 0.72m | 569.1 (23.21%) | 1.02m | 739.5 (10.51%) | 1.75m | 922.9 (-1.64%) | 2.53m |
| | Ours | **369.2 (13.18%)** | 0.13m | **624.8 (4.53%)** | 0.28m | **531.4 (41.00%)** | 0.55m | **559.9 (21.22%)** | 0.81m | **732.6 (9.47%)** | 1.58m | **920.1 (-1.94%)** | 2.55m |
| Sampling | DANIEL | 366.7 (12.42%) | 0.88s | 629.9 (5.39%) | 1.84s | 521.8 (38.45%) | 3.83s | 552.6 (19.64%) | 5.97s | 725.3 (8.38%) | 12.17s | 914.0 (-2.59%) | 21.09s |
| | AS | 356.1 (9.17%) | 0.28m | 620.7 (3.85%) | 0.63m | 502.6 (33.35%) | 1.17m | 537.1 (16.28%) | 1.78m | 712.8 (6.52%) | 3.38m | 902.8 (-3.78%) | 5.03m |
| | EAS | 354.9 (8.80%) | 0.26m | 619.4 (3.64%) | 0.53m | 500.5 (32.79%) | 1.03m | 540.8 (17.08%) | 1.62m | 714.9 (6.83%) | 2.92m | 903.4 (-3.72%) | 4.67m |
| | Ours | **347.7 (6.59%)** | 0.17m | **607.9 (1.71%)** | 0.38m | **499.2 (32.45%)** | 0.83m | **529.9 (14.72%)** | 1.30m | **703.3 (5.10%)** | 2.67m | **889.5 (-5.20%)** | 4.52m |

Table 2: Generalization performance evaluation on public benchmark datasets.

| | | mk Obj. (Gap) | Time | la (rdata) Obj. (Gap) | Time | la (edata) Obj. (Gap) | Time | la (vdata) Obj. (Gap) | Time |
|---|---|---|---|---|---|---|---|---|---|
| | Ortools | 173.9 (0.00%) | 0.51h | 933.4 (0.00%) | 0.71h | 1026.9 (0.00%) | 5.41m | 920.6 (0.00%) | 0.75h |
| | MWKR | 202.2 (16.27%) | 0.49s | 1052.8 (12.79%) | 0.52s | 1218.8 (18.69%) | 0.52s | 952.0 (3.41%) | 0.52s |
| SD1 10 × 5 model Greedy | DANIEL | 185.7 (6.79%) | 1.29s | 1031.6 (10.52%) | 1.37s | 1194.9 (16.36%) | 1.36s | 944.9 (2.64%) | 1.37s |
| | AS | **182.6 (5.00%)** | 0.85m | 1008.2 (8.01%) | 0.90m | 1159.7 (12.93%) | 0.91m | 936.7 (1.75%) | 0.91m |
| | EAS | **182.7 (5.06%)** | 0.73m | 992.0 (6.28%) | 0.80m | 1139.9 (11.00%) | 0.79m | 930.3 (1.05%) | 0.79m |
| | Ours | **182.3 (4.83%)** | 0.61m | **982.8 (5.29%)** | 0.67m | **1120.8 (9.14%)** | 0.68m | **928.1 (0.81%)** | 0.67m |
| Sampling | DANIEL | 180.8 (3.97%) | 4.13s | 978.3 (4.97%) | 4.71s | 1122.6 (9.25%) | 4.73s | 925.4 (0.53%) | 4.77s |
| | AS | 179.2 (3.15%) | 1.30m | 970.5 (4.00%) | 1.42m | 1107.6 (8.60%) | 1.40m | 923.0 (0.25%) | 1.41m |
| | EAS | 179.3 (3.15%) | 1.20m | 969.5 (4.11%) | 1.29m | 1101.8 (7.32%) | 1.38m | 922.7 (0.23%) | 1.32m |
| | Ours | **177.0 (2.25%)** | 0.95m | **963.9 (3.28%)** | 1.13m | **1086.0 (5.75%)** | 1.13m | **921.9 (0.14%)** | 1.07m |
| SD1 15 × 10 model Greedy | DANIEL | 184.4 (6.06%) | 1.30s | 1040.0 (11.39%) | 1.36s | 1175.5 (14.88%) | 1.38s | 948.7 (3.05%) | 1.37s |
| | AS | **182.4 (5.02%)** | 0.85m | 1014.5 (8.63%) | 0.89m | 1152.9 (12.30%) | 0.93m | 934.8 (2.29%) | 0.91m |
| | EAS | **182.2 (5.01%)** | 0.73m | 1004.1 (7.57%) | 0.81m | 1140.0 (11.02%) | 0.79m | 932.0 (1.24%) | 0.79m |
| | Ours | **181.5 (4.38%)** | 0.61m | **984.9 (5.54%)** | 0.67m | **1131.0 (10.18%)** | 0.67m | **928.7 (0.88%)** | 0.67m |
| Sampling | DANIEL | 180.9 (3.99%) | 4.08s | 983.3 (5.35%) | 4.73s | 1119.7 (8.73%) | 4.70s | 925.7 (0.55%) | 4.75s |
| | AS | 178.9 (2.91%) | 1.31m | 971.2 (4.25%) | 1.42m | 1109.1 (8.97%) | 1.42m | 922.9 (0.24%) | 1.40m |
| | EAS | 178.7 (2.85%) | 1.22m | 971.6 (4.29%) | 1.29m | 1106.5 (8.74%) | 1.36m | 922.5 (0.22%) | 1.32m |
| | Ours | **177.4 (2.03%)** | 0.95m | **967.3 (3.64%)** | 1.13m | **1094.1 (7.31%)** | 1.15m | **921.7 (0.12%)** | 1.07m |

we implement EAS-EMB for comparison due to its superior performance in scheduling problems as reported in (Hottung et al., 2021b). For fair comparison, all fine-tuning methods (AS, EAS and ours) are performed with a fixed number of $K = 10$ adaptation steps per test instance. We use the solutions generated by OR-Tools as reference to compute the objective gap of each method.

## 5.2 PERFORMANCE EVALUATION

**Results on synthetic data.** Table 1 reports the average makespan, relative optimality gap, and average run time for solving an instance across all groups. Note that to assess the generalization ability to larger problem sizes, we solve the unseen $30 \times 10$ and $40 \times 10$ instances using models trained on the closest scale, $20 \times 10$. For all neural methods, we report their performance under the greedy and sampling mode (100 solutions as in (Song et al., 2023; Wang et al., 2023)). For our method, we use MC averaging and MC best-sample as the inner-loop gradient estimator for the greedy and sampling mode, respectively. Clearly, all neural methods significantly outperform the MWKR rule. AS and EAS can improve the original DANIEL model by test-time adaptation, and the latter is better in most cases especially on SD2. Our method consistently outperforms all baselines across all settings, because the meta-learning scheme explicitly considers the fine-tuning process and the trained meta-model provides a good start point for instance-wise adaptation. Moreover, benefiting from its gradient-free nature, our method allows full-parameter adaptation without backpropagation, achieving superior efficiency and surpasses EAS in terms of both solution quality and run time. In Appendix B, we provide results on much larger $50 \times 20$ and $100 \times 20$ instances,

Table 3: Effectiveness of derivative-free meta-learning.

|  | $10 \times 5$ | $20 \times 5$ | $15 \times 10$ | $20 \times 10$ |
|---|---|---|---|---|
| DANIEL | 25.20% | 12.27% | 56.86% | 32.09% |
| AS | 20.17% | 7.87% | 47.97% | 23.73% |
| EAS | 16.65% | 7.22% | 45.32% | 23.21% |
| ES | 14.38% | 5.02% | 44.81% | 24.20% |
| Ours | **13.18%** | **4.54%** | **41.00%** | **21.22%** |

Table 4: Inference time of GPU and CPU implementation.

| Instance | GPU (Ours) | CPU (Ray) |
|---|---|---|
| $10 \times 5$ | 0.5s | 9.8s |
| $20 \times 5$ | 1.1s | 29.8s |
| $15 \times 10$ | 2.4s | 62.1s |
| $20 \times 10$ | 3.8s | 122.3s |

Table 5: Comparison with FOMAML (-G: greedy; -S: sampling).

|  | $10 \times 5$ | | $20 \times 5$ | | $15 \times 10$ | | $20 \times 10$ | | $30 \times 10^*$ | | $40 \times 10^*$ | |
|---|---|---|---|---|---|---|---|---|---|---|---|---|
|  | Obj. (Gap) | Time | Obj. (Gap) | Time | Obj. (Gap) | Time | Obj. (Gap) | Time | Obj. (Gap) | Time | Obj. (Gap) | Time |
| FOMAML-G | 394.4 (20.91%) | 0.14m | 646.6 (8.18%) | 0.30m | 557.1 (47.83%) | 0.51m | 583.8 (26.39%) | 0.71m | 757.6 (13.21%) | 1.13m | 949.1 (1.17%) | 1.63m |
| Ours-G | **369.2 (13.18%)** | 0.13m | **624.8 (4.53%)** | 0.28m | **531.4 (41.00%)** | 0.55m | **559.9 (21.22%)** | 0.81m | **732.6 (9.47%)** | 1.58m | **920.1 (-1.94%)** | 2.55m |
| FOMAML-S | 356.0 (9.13%) | 0.16m | 625.2 (4.61%) | 0.37m | 514.0 (36.36%) | 0.68m | 551.4 (19.36%) | 1.01m | 733.3 (9.58%) | 1.83m | 928.7 (-1.02%) | 2.95m |
| Ours-S | **347.7 (6.59%)** | 0.17m | **607.9 (1.71%)** | 0.38m | **499.2 (32.45%)** | 0.83m | **529.9 (14.72%)** | 1.30m | **703.3 (5.10%)** | 2.67m | **889.5 (-5.20%)** | 4.52m |

**Results on public benchmark instances.** We further evaluate the cross-distribution generalization ability of our method on the widely used public benchmarks. We follow (Wang et al., 2023) and use the meta-models trained on SD1 with sizes $10 \times 5$ and $15 \times 10$ for evaluation. As shown in Table 2, our method consistently demonstrates strong performance as observed on the synthetic data. It outperforms all baseline methods across all settings, with particularly notable improvements on the edata task, highlighting its robust generalization ability in handling out-of-distribution instances. Further statistical analysis can be found in Appendix E.

## 5.3 ANALYSES

**Effectiveness of derivative-free meta-learning.** Here we perform a more fine-grained analysis on the two major parts of our method, i.e. the DFO method and the meta-learning framework. Specifically, on SD2, we train and fine-tune the DANIEL model using the same DFO procedure as in our method, namely ES. As shown in Table 3, this method can already surpass existing gradient-based training and fine-tuning methods, showing the effectiveness of DFO. Building upon this, the integration of meta-learning further enhances the overall fine-tuning performance.

**Comparison with standard meta-learning.** We present a direct comparison between the instance-wise adaptive meta-learning framework proposed in this work and the standard gradient-based meta-learning method, FOMAML, on the challenging SD2 dataset. All experiments were conducted under identical conditions, including the same model architecture, initialization, training epochs (200) and fine-tuning steps (10) to ensure fairness. As shown in Table 5, under the same setting, our method consistently outperforms FOMAML across all instance scales with comparable inference speed. Training cost comparison with FOMAML is presented in Appendix C due to the space limit.

**Analysis of inner-loop gradient estimators.** Next, we verify the effectiveness of our two inner-loop gradient estimators proposed in Section 4.1.1, MC averaging and MC best-sample. The evaluation is conducted on $10 \times 5$ instances from SD2, and the standard NES implementation (the single-sample estimator), AS and EAS are also incorporated for reference. All methods are evaluated under the sampling mode. In Figure 3, we plot the average objective value for each of the $K = 10$ fine-tuning steps. As shown in Figure 3, the single-sample strategy performs only on par with AS and EAS. With our two MC strategies, performance of the meta-model is significantly boosted, and MC best-sample demonstrated the strongest adaptation ability, showing the effectiveness of our novel design.

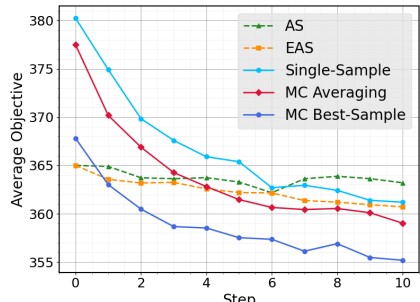

Figure 3: Comparison of different inner-loop gradient estimators.

**Analysis of inference efficiency.** We compare the efficiency of our GPU-based implementation with the Ray-based CPU multithreading strategy commonly used in existing DFO methods. Specifically, we measure the run time required to complete one forward pass for evaluating the fitness of 100 individuals. The Ray-based implementation utilizes all 16 available threads on our machine to maximize CPU parallelism. As shown in Table 4, our GPU implementation is much faster than the CPU-based method across various scales. Furthermore, as the problem scale increases, the advantage of our implementation becomes even more prominent.

## 6  CONCLUSION AND FUTURE WORK

While deep reinforcement learning has been successfully applied in complex scheduling problems, the per-instance solving performance is still far from optimality. In this paper, we propose a derivative-free meta-learning framework to enhance the ability of learning-based scheduling models in adapting to individual instances at test time. The trained meta-model provides a good start point for instance-wise fine-tuning, and the strong empirical performance of our method is validated on both FJSP (reinforcement learning) and JSP (self-supervised learning). One limitation of our method is that derivative-free optimization is known to be relatively slower than gradient-based methods. In the future, we will investigate more effective ways to improve training efficiency.

### ACKNOWLEDGMENTS

This research was supported by the National Natural Science Foundation of China under Grants 62473233, U23B2059, and the Guangdong Basic and Applied Basic Research Foundation under Grant 2025A1515011704.

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

Table 6: SD1 instance generation distributions.

| Size($n \times m$) | $|O_i|$ | $|M_{ij}|$ | $\bar{p}_{ij}$ |
|---|---|---|---|
| $10 \times 5$ | U(4, 6) | U(1, 5) | U(1, 20) |
| $20 \times 5$ | U(4, 6) | U(1, 5) | U(1, 20) |
| $15 \times 10$ | U(8, 12) | U(1, 10) | U(1, 20) |
| $20 \times 10$ | U(8, 12) | U(1, 10) | U(1, 20) |
| $30 \times 10$ | U(8, 12) | U(1, 10) | U(1, 20) |
| $40 \times 10$ | U(8, 12) | U(1, 10) | U(1, 20) |

Table 7: SD2 instance generation distributions.

| Size($n \times m$) | $|O_i|$ | $|M_{ij}|$ | $p_{ijk}$ |
|---|---|---|---|
| $10 \times 5$ | 5 | U(1, 5) | U(1, 99) |
| $20 \times 5$ | 5 | U(1, 5) | U(1, 99) |
| $15 \times 10$ | 10 | U(1, 10) | U(1, 99) |
| $20 \times 10$ | 10 | U(1, 10) | U(1, 99) |
| $30 \times 10$ | 10 | U(1, 10) | U(1, 99) |
| $40 \times 10$ | 10 | U(1, 10) | U(1, 99) |

Table 8: Generalization to very large FJSP instances. For neural methods, values outside (inside) parenthesis are greedy (sampling) results.

| | $50 \times 20$ | $100 \times 20$ |
|---|---|---|
| Ortools | 972.8 | 1737.1 |
| DANIEL | 1013.9 (959.6) | 1649.5 (1612.9) |
| AS | 960.4 (930.7) | OOM |
| EAS | 970.1 (942.3) | OOM |
| Ours (Zero-shot) | 949.2 (904.9) | 1609.1 (1562.2) |
| Ours (Fine-tune) | **886.5 (875.4)** | **1510.5 (1492.2)** |

## A    FJSP DATASETS

In this study, we consider six synthetic instance sizes for training and testing the FJSP. For the SD1 dataset, the instance generation process follows the classical method proposed in (Brandimarte, 1993). Specifically, for each instance size, the number of operations $|O_i|$ in each job $J_i$, the number of compatible machines $|M_{ij}|$ for each operation $O_{ij}$, and the average processing time $\bar{p}_{ij}$ of each operation across its compatible machines are all independently sampled from uniform distributions defined in Table 6. Then, the actual processing time $p_{ijk}$ of operation $o_{ij}$ on a specific compatible machine $M_k \in M_{ij}$ is sampled from a bounded uniform distribution centered around the average processing time, i.e., $p_{ijk} \sim U(0.8\bar{p}_{ij}, 1.2\bar{p}_{ij})$. In contrast, for the SD2 dataset, the number of operations in each job is set equal to the total number of machines in the shop. The processing time $p_{ijk}$ of each operation on each compatible machine is directly sampled from a wider uniform distribution, resulting in greater variability in operation durations. The detailed parameter settings are provided in Table 7.

## B    GENERALIZATION TO LARGE PROBLEMS

To further examine scalability at inference time, we conducted an experiment on very large FJSP instances of sizes $50 \times 20$ and $100 \times 20$ using SD2 distribution. For our method and DANIEL, we use the model trained on $20 \times 10$ instances. As shown in Table 8, our meta-model shows strong generalization performance. It significantly outperforms DANIEL trained on the same size in both greedy and sampling modes. Its zero-shot performance already exceeds the 1-hour results of Ortools, and fine-tuning further boosts the performance. Notably, AS/EAS failed on the largest 100×20 instances (out-of-memory), since their gradient-based fine-tuning requires large GPU memory to store gradients. In contrast, our gradient-free approach is much more memory friendly on these very large problems.

## C    TRAINING COST COMPARISON

To assess the training efficiency of our method, we provide a detailed comparison of the training cost against the FOMAML baseline, as shown in Table 9. The comparison considers training time (in GPU hours) and GPU memory usage over 200 epochs. Our method incurs approximately 1.5-2 times higher training time and GPU memory usage compared to FOMAML. This increase is attributed to the nature of ES, which performs a population-based search and evaluates multiple perturbations during each update. Nevertheless, we believe this additional training cost is modest and reasonable,

Table 9: Training Cost Comparison with FOMAML on SD2.

|  |  | $10 \times 5$ | $20 \times 5$ | $15 \times 10$ | $20 \times 10$ |
|---|---|---|---|---|---|
| Ours | Training Time | 2.1h | 4.5h | 7.2h | 11.2h |
|  | GPU Memory Usage | 1.3G | 1.6G | 2.0G | 2.2G |
| FOMAML | Training Time | 1.5h | 2.9h | 3.7h | 5.5h |
|  | GPU Memory Usage | 0.9G | 1.0G | 1.0G | 1.2G |

Table 10: JSP Experiments on Taillard's benchmarks. Instance sizes marked with $^*$ were not seen during training.

|  |  | $15 \times 15$ |  | $20 \times 15$ |  | $20 \times 20$ |  | $30 \times 15^*$ |  | $30 \times 20^*$ |  |
|---|---|---|---|---|---|---|---|---|---|---|---|
|  |  | Gap | Time | Gap | Time | Gap | Time | Gap | Time | Gap | Time |
| Greedy | SPN | 16.86% | 0.47s | 16.13% | 0.67s | 19.01% | 0.87s | 21.16% | 1.06s | 22.03% | 1.43s |
|  | AS | 11.87% | 0.21m | 14.00% | 0.28m | 14.00% | 0.39m | 16.76% | 0.43m | 17.87% | 0.58m |
|  | EAS | 11.94% | 0.16m | 13.26% | 0.23m | 13.98% | 0.28m | 16.10% | 0.32m | 17.76% | 0.43m |
|  | Ours | **10.83%** | 0.13m | **12.73%** | 0.19m | **11.78%** | 0.26m | **15.08%** | 0.32m | **17.08%** | 0.42m |
| Sampling | SPN | 8.52% | 0.53s | 10.31% | 0.71s | 11.51% | 0.96s | 13.57% | 1.20s | 15.91% | 1.57s |
|  | AS | 7.54% | 0.23m | 9.24% | 0.29m | 9.43% | 0.38m | 12.38% | 0.43m | 14.52% | 0.57m |
|  | EAS | 7.06% | 0.17m | 9.28% | 0.24m | 9.84% | 0.30m | 12.21% | 0.33m | 14.33% | 0.45m |
|  | Ours | **6.31%** | 0.15m | **8.76%** | 0.21m | **9.24%** | 0.28m | **11.80%** | 0.33m | **14.07%** | 0.43m |

since training is offline and the final solution quality during inference time is significantly better than the baselines.

# D  JSP EXPERIMENTS

In this section, we apply our method to a self-supervised learning model for JSP, the Self-labeling Pointer Network (SPN) (Corsini et al., 2024), to evaluate its effectiveness under a different learning paradigm and model to further validate its generality and robustness. SPN is a self-supervised neural scheduling framework tailored for JSP. A key advantage of SPN is that it eliminates the need for external supervision or reinforcement learning signals. During training, SPN utilizes a Pointer Network to construct scheduling solutions by sampling multiple candidate sequences for each instance. It then selects the one with the lowest makespan as a pseudo-label and optimizes the model via cross-entropy loss. This iterative process progressively improves the scheduling quality without relying on ground-truth labels.

**Dataset.** To ensure a fair comparison, we train both SPN and our meta-model on the same dataset. Following the protocol in (Corsini et al., 2024), we adopt a fixed training dataset composed of six instance sizes ($n \times m$): {$10 \times 10$, $15 \times 10$, $15 \times 15$, $20 \times 10$, $20 \times 15$, $20 \times 20$}, with 500 randomly generated instances per size, totaling 3,000 training instances per epoch. For evaluation, we assess the generalization ability of the models using five instance sizes selected from Taillard's benchmark set (Taillard, 1993), ranging from $15 \times 15$ to $30 \times 20$.

**Training Setup.** For our method, the population size is set to $\mu = 100$ and the noise standard deviation to $\sigma = 0.04$. The step sizes for both the inner and outer loops are set to $\alpha = \beta = 5 \times 10^{-3}$. The batch size is fixed to $B = 10$, consistent with the SPN training configuration. For each instance, we perform $K = 3$ inner-loop adaptation steps. Given the fast convergence behavior of our method, we train for a total of 10 epochs, while the SPN baseline is trained for 20 epochs as in the original paper. During the inner-loop adaptation phase, we set the number of parallel samples $L$ for the two proposed gradient estimators as follows: $L = 20$ for the MC averaging, and $L = 128$ for the MC best-sample, which aligns with the number of sampled solutions $\beta$ used in SPN training.

**Performance Evaluation.** We compare our method against the original SPN model as well as the two test-time fine-tuning strategies, AS and EAS. All fine-tuning methods, including AS, EAS, and ours, perform a fixed number of $K = 10$ adaptation steps on each test instance. To evaluate

Table 11: Average percentage gaps and standard deviations (mean ± std) on synthetic test sets. Instance sizes marked with * were not seen during training.

| | | | $10 \times 5$ | $20 \times 5$ | $15 \times 10$ | $20 \times 10$ | $30 \times 10^*$ | $40 \times 10^*$ |
|---|---|---|---|---|---|---|---|---|
| SD1 | Greedy | DANIEL | $10.80\% \pm 5.59\%$ | $4.94\% \pm 1.90\%$ | $10.48\% \pm 3.91\%$ | $1.17\% \pm 2.04\%$ | $2.07\% \pm 1.54\%$ | $1.17\% \pm 1.45\%$ |
| | | AS | $8.93\% \pm 3.13\%$ | $2.97\% \pm 0.91\%$ | $7.47\% \pm 3.38\%$ | $-0.76\% \pm 1.27\%$ | $0.94\% \pm 1.27\%$ | $0.38\% \pm 1.28\%$ |
| | | EAS | $7.68\% \pm 2.38\%$ | $3.03\% \pm 1.00\%$ | $7.12\% \pm 2.98\%$ | $-0.87\% \pm 1.37\%$ | $0.98\% \pm 1.30\%$ | $0.33\% \pm 1.24\%$ |
| | | Ours | $\mathbf{7.06\% \pm 1.88\%}$ | $\mathbf{1.01\% \pm 0.62\%}$ | $\mathbf{5.21\% \pm 2.55\%}$ | $\mathbf{-2.09\% \pm 1.12\%}$ | $\mathbf{-0.18\% \pm 1.19\%}$ | $\mathbf{-0.71\% \pm 1.22\%}$ |
| | Sampling | DANIEL | $5.61\% \pm 2.01\%$ | $2.39\% \pm 0.74\%$ | $4.93\% \pm 1.92\%$ | $-1.17\% \pm 1.00\%$ | $1.23\% \pm 1.25\%$ | $0.90\% \pm 1.21\%$ |
| | | AS | $4.36\% \pm 1.27\%$ | $1.70\% \pm 0.59\%$ | $3.77\% \pm 1.76\%$ | $-1.89\% \pm 0.94\%$ | $0.76\% \pm 1.22\%$ | $0.35\% \pm 1.23\%$ |
| | | EAS | $4.15\% \pm 1.26\%$ | $1.91\% \pm 0.59\%$ | $3.70\% \pm 1.75\%$ | $-1.89\% \pm 0.92\%$ | $0.69\% \pm 1.20\%$ | $0.35\% \pm 1.38\%$ |
| | | Ours | $\mathbf{3.32\% \pm 0.95\%}$ | $\mathbf{0.16\% \pm 0.41\%}$ | $\mathbf{2.05\% \pm 1.93\%}$ | $\mathbf{-3.62\% \pm 0.94\%}$ | $\mathbf{-0.87\% \pm 1.18\%}$ | $\mathbf{-1.14\% \pm 1.25\%}$ |
| SD2 | Greedy | DANIEL | $25.20\% \pm 9.09\%$ | $12.27\% \pm 4.49\%$ | $56.86\% \pm 11.63\%$ | $32.09\% \pm 8.06\%$ | $15.75\% \pm 5.51\%$ | $2.59\% \pm 4.25\%$ |
| | | AS | $20.17\% \pm 7.07\%$ | $7.87\% \pm 3.06\%$ | $47.97\% \pm 8.36\%$ | $23.73\% \pm 4.77\%$ | $10.22\% \pm 3.69\%$ | $-1.14\% \pm 3.11\%$ |
| | | EAS | $16.65\% \pm 5.80\%$ | $7.22\% \pm 2.73\%$ | $45.32\% \pm 8.15\%$ | $23.21\% \pm 4.49\%$ | $10.51\% \pm 3.01\%$ | $-1.64\% \pm 2.94\%$ |
| | | Ours | $\mathbf{13.18\% \pm 4.75\%}$ | $\mathbf{4.53\% \pm 1.79\%}$ | $\mathbf{41.00\% \pm 6.60\%}$ | $\mathbf{21.22\% \pm 3.99\%}$ | $\mathbf{9.47\% \pm 2.87\%}$ | $\mathbf{-1.94\% \pm 2.86\%}$ |
| | Sampling | DANIEL | $12.42\% \pm 4.19\%$ | $5.39\% \pm 2.06\%$ | $38.45\% \pm 6.30\%$ | $19.64\% \pm 3.34\%$ | $8.38\% \pm 2.98\%$ | $-2.59\% \pm 2.84\%$ |
| | | AS | $9.17\% \pm 3.42\%$ | $3.85\% \pm 1.26\%$ | $33.35\% \pm 5.68\%$ | $16.28\% \pm 2.60\%$ | $6.52\% \pm 2.51\%$ | $-3.78\% \pm 2.66\%$ |
| | | EAS | $8.80\% \pm 3.62\%$ | $3.64\% \pm 1.32\%$ | $32.79\% \pm 5.30\%$ | $17.08\% \pm 2.86\%$ | $6.83\% \pm 2.36\%$ | $-3.72\% \pm 2.67\%$ |
| | | Ours | $\mathbf{6.59\% \pm 2.61\%}$ | $\mathbf{1.71\% \pm 0.99\%}$ | $\mathbf{32.45\% \pm 5.08\%}$ | $\mathbf{14.72\% \pm 3.08\%}$ | $\mathbf{5.10\% \pm 2.01\%}$ | $\mathbf{-5.20\% \pm 2.57\%}$ |

Table 12: Average percentage gaps and standard deviations (mean ± std) on public benchmarks.

| | | | mk Gap | la (rdata) Gap | la (edata) Gap | la (vdata) Gap |
|---|---|---|---|---|---|---|
| SD1 $10 \times 5$ model | Greedy | DANIEL | $6.79\% \pm 3.49\%$ | $10.52\% \pm 8.20\%$ | $16.36\% \pm 5.62\%$ | $2.64\% \pm 3.02\%$ |
| | | AS | $5.00\% \pm 2.84\%$ | $8.01\% \pm 5.93\%$ | $12.93\% \pm 5.65\%$ | $1.75\% \pm 2.63\%$ |
| | | EAS | $5.06\% \pm 2.26\%$ | $6.28\% \pm 5.85\%$ | $11.00\% \pm 5.48\%$ | $1.05\% \pm 1.01\%$ |
| | | Ours | $\mathbf{4.83\% \pm 1.52\%}$ | $\mathbf{5.29\% \pm 4.67\%}$ | $\mathbf{9.14\% \pm 5.09\%}$ | $\mathbf{0.81\% \pm 0.84\%}$ |
| | Sampling | DANIEL | $3.97\% \pm 1.34\%$ | $4.97\% \pm 4.56\%$ | $9.25\% \pm 4.51\%$ | $0.53\% \pm 0.56\%$ |
| | | AS | $3.15\% \pm 0.30\%$ | $4.00\% \pm 4.12\%$ | $8.60\% \pm 4.49\%$ | $0.25\% \pm 0.49\%$ |
| | | EAS | $3.15\% \pm 0.44\%$ | $4.11\% \pm 3.93\%$ | $7.32\% \pm 4.26\%$ | $0.23\% \pm 0.38\%$ |
| | | Ours | $\mathbf{2.25\% \pm 0.67\%}$ | $\mathbf{3.28\% \pm 3.65\%}$ | $\mathbf{5.75\% \pm 3.96\%}$ | $\mathbf{0.14\% \pm 0.37\%}$ |
| SD1 $15 \times 10$ model | Greedy | DANIEL | $6.06\% \pm 2.43\%$ | $11.39\% \pm 7.72\%$ | $14.88\% \pm 5.41\%$ | $3.05\% \pm 3.17\%$ |
| | | AS | $5.02\% \pm 1.85\%$ | $8.63\% \pm 7.10\%$ | $12.30\% \pm 5.24\%$ | $2.29\% \pm 2.07\%$ |
| | | EAS | $5.01\% \pm 1.83\%$ | $7.57\% \pm 5.54\%$ | $11.02\% \pm 4.97\%$ | $1.24\% \pm 1.93\%$ |
| | | Ours | $\mathbf{4.38\% \pm 1.46\%}$ | $\mathbf{5.54\% \pm 4.87\%}$ | $\mathbf{10.18\% \pm 5.20\%}$ | $\mathbf{0.88\% \pm 1.09\%}$ |
| | Sampling | DANIEL | $3.99\% \pm 1.62\%$ | $5.35\% \pm 4.12\%$ | $8.73\% \pm 4.48\%$ | $0.55\% \pm 0.61\%$ |
| | | AS | $2.91\% \pm 0.59\%$ | $4.25\% \pm 3.64\%$ | $8.97\% \pm 4.49\%$ | $0.24\% \pm 0.50\%$ |
| | | EAS | $2.85\% \pm 1.19\%$ | $4.29\% \pm 3.68\%$ | $8.74\% \pm 4.30\%$ | $0.22\% \pm 0.45\%$ |
| | | Ours | $\mathbf{2.03\% \pm 0.66\%}$ | $\mathbf{3.64\% \pm 3.63\%}$ | $\mathbf{7.31\% \pm 4.90\%}$ | $\mathbf{0.12\% \pm 0.44\%}$ |

Table 13: Average percentage gaps and standard deviations (mean ± std) on Taillard's benchmarks. Instance sizes marked with * were not seen during training.

| | | $15 \times 15$ | $20 \times 15$ | $20 \times 20$ | $30 \times 15^*$ | $30 \times 20^*$ |
|---|---|---|---|---|---|---|
| Greedy | SPN | $16.86\% \pm 2.75\%$ | $16.12\% \pm 3.55\%$ | $19.01\% \pm 3.26\%$ | $21.15\% \pm 4.84\%$ | $22.03\% \pm 2.89\%$ |
| | AS | $11.86\% \pm 1.94\%$ | $13.99\% \pm 2.54\%$ | $14.00\% \pm 2.17\%$ | $16.76\% \pm 3.46\%$ | $17.87\% \pm 2.64\%$ |
| | EAS | $11.94\% \pm 1.34\%$ | $13.26\% \pm 2.43\%$ | $13.97\% \pm 1.96\%$ | $16.10\% \pm 2.74\%$ | $17.76\% \pm 2.13\%$ |
| | Ours | $\mathbf{10.83\% \pm 2.42\%}$ | $\mathbf{12.73\% \pm 1.50\%}$ | $\mathbf{11.78\% \pm 2.00\%}$ | $\mathbf{15.08\% \pm 3.56\%}$ | $\mathbf{17.08\% \pm 1.51\%}$ |
| Sampling | SPN | $8.52\% \pm 1.99\%$ | $10.31\% \pm 2.13\%$ | $11.50\% \pm 1.16\%$ | $13.57\% \pm 2.90\%$ | $15.91\% \pm 1.62\%$ |
| | AS | $7.53\% \pm 1.44\%$ | $9.23\% \pm 1.73\%$ | $9.43\% \pm 1.18\%$ | $12.38\% \pm 2.92\%$ | $14.52\% \pm 1.81\%$ |
| | EAS | $7.06\% \pm 1.77\%$ | $9.28\% \pm 1.67\%$ | $9.84\% \pm 1.05\%$ | $12.21\% \pm 2.92\%$ | $14.33\% \pm 1.50\%$ |
| | Ours | $\mathbf{6.31\% \pm 2.01\%}$ | $\mathbf{8.75\% \pm 1.10\%}$ | $\mathbf{9.23\% \pm 1.40\%}$ | $\mathbf{11.80\% \pm 2.87\%}$ | $\mathbf{14.07\% \pm 1.56\%}$ |

performance, we use the best solution value of each instance as a reference and report the objective gap accordingly. Table 10 summarizes the results of all methods under both greedy and sampling modes, where the sampling mode generates 128 candidate solutions per instance following Corsini et al. (2024). As shown in the table, our method consistently outperforms all baselines across both evaluation settings, even when applied in this non-reinforcement learning paradigm. These results highlight the effectiveness, generality, and model-agnostic nature of our approach.

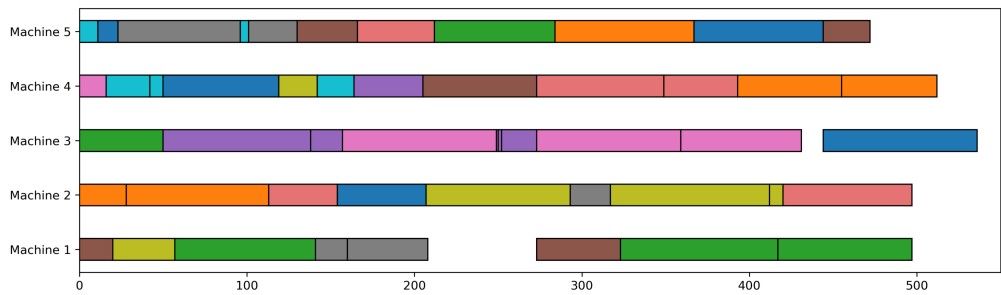

Figure 4: FJSP Gantt Chart with Makespan of 536 (Before Fine-Tuning)

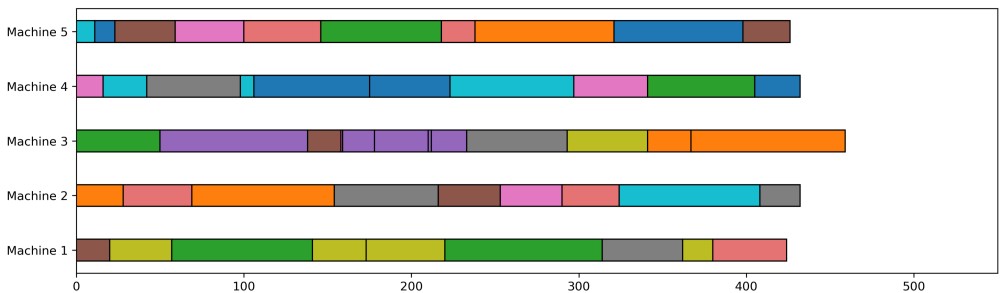

Figure 5: FJSP Gantt Chart with Makespan of 459 (After Fine-Tuning)

## E    STATISTICAL RESULTS

Tables 11 , 12 and 13 present the average optimality gaps and standard deviations of all neural-based methods across various instance sizes on the synthetic and public benchmark datasets, respectively. These results confirm that our conclusions remain valid even when accounting for variance. Our meta-learning framework explicitly incorporates the fine-tuning process during training, enabling it to provide well-initialized parameters for each test instance. Moreover, by leveraging the gradient-free nature of our optimization scheme, the method facilitates high-quality instance-wise adaptation. Across all settings, our approach consistently outperforms all baseline methods.

## F    VISUALIZATION OF DECISION ADAPTATION IN FJSP

To illustrate the adaptation process of scheduling decisions in the FJSP, we selected a representative 10x5 FJSP instance and used Gantt charts to depict the changes in decision-making before and after model fine-tuning. In the Gantt charts, blocks of the same color represent the operations of the same workpiece, with the horizontal axis representing processing time and the vertical axis representing different machines. Figure 4 shows the scheduling decisions before model fine-tuning. Since the model was not optimized for a specific instance, it resulted in extended idle times for machines, and the completion times of different machines varied significantly. After fine-tuning the model (as shown in Figure 5), idle times were eliminated, and the completion times across different machines became more consistent, leading to a significant reduction in makespan.

