# OpenReview forum: "Instance-wise Adaptive Scheduling via Derivative-Free Meta-Learning"
_ICLR.cc/2026/Conference — ICLR 2026 Poster_

### Official Review · Reviewer_kVaC · 2025-10-16

**Soundness:** 2
**Presentation:** 2
**Contribution:** 2
**Rating:** 2
**Confidence:** 4

**Summary:**

This paper proposes Instance-wise Adaptive Scheduling via Derivative-Free Meta-Learning, a framework that improves deep reinforcement learning for complex scheduling problems such as the Flexible Job-Shop Scheduling Problem (FJSP). It uses meta-learning to learn a generalizable initialization for fast per-instance adaptation and replaces gradient-based optimization with a derivative-free method based on Evolution Strategies, enhanced by GPU parallelization for efficiency. Experiments show that the approach consistently outperforms existing baselines like Active Search and Efficient Active Search in solution quality, adaptability, and generalization.

**Strengths:**

The paper’s key strengths lie in its novel integration of meta-learning with derivative-free optimization to improve instance-wise adaptability in complex scheduling tasks. By leveraging Evolution Strategies and a GPU-parallelized implementation, the approach achieves efficient, gradient-free training and fine-tuning that reportedly scales well to large problem instances.

**Weaknesses:**

While the paper demonstrates solid implementation and empirical results, its conceptual novelty is limited. The proposed framework largely repurposes the MAML paradigm with minor modifications, and the use of derivative-free optimization appears incremental rather than innovative. The motivation for replacing gradient-based methods is not well justified, with very limited theoretical or empirical evidence. Consequently, the work feels more like an engineering improvement than a substantial methodological advance, lacking deeper analysis or insight into why the proposed approach is fundamentally superior.

A key weakness appears to be the use of two nested evolutionary processes, one for instance-level adaptation and another for meta-level optimization. They make the method highly computationally expensive and sample inefficient. Since evolutionary algorithms already demand large populations and many evaluations, nesting them greatly amplifies the cost. Although GPU parallelization helps with runtime, it does not address this fundamental inefficiency. The paper also lacks analysis showing that such a two-level structure is necessary, raising concerns about scalability and practical applicability.

As acknowledged by the authors, a potential advantage of the proposed nested design is that it avoids computing second-order derivatives, which can be costly in traditional MAML frameworks. However, it remains unclear why ES is the preferred or necessary choice for achieving this goal. There exist several other first-order or gradient-approximation approaches that could eliminate second-order terms with far lower computational cost. The paper does not provide justification or comparative analysis to show that ES offers a clear advantage over these alternatives, making the design choice appear arbitrary and potentially inefficient.

A significant flaw lies in the paper’s misuse of the term “natural gradient.” Although the authors claim to adopt Natural Evolution Strategies (NES), their algorithm actually follows the OpenAI-ES formulation, which estimates a standard Monte Carlo gradient without the Fisher preconditioning required for true NES. In particular, Equation (4) is mathematically identical to the OpenAI-ES update rule, showing no theoretical modification or geometric correction. This indicates a misunderstanding of NES and renders the claim of “natural gradient estimation” technically inaccurate, undermining both the paper’s methodological novelty and its conceptual soundness.

The claim that “the entire fitness evaluation process is offloaded to GPU” in the paper appears overstated. In practice, the feasibility of full GPU offloading strongly depends on the problem structure and the simulator used for fitness evaluation. Many scheduling environments, particularly those involving discrete event simulation or complex resource constraints, are not inherently GPU-friendly and may still require significant CPU-side computation or data transfer. Without demonstrating that the underlying environment is fully GPU-parallelizable, this claim lacks general validity. As a result, the reported efficiency gains may not generalize beyond the specific experimental setup used in the paper.

While the experimental results in the paper show consistent improvements over baselines such as Active Search and Efficient Active Search, the gains are generally incremental rather than dramatic. The improvements in optimality gap are often modest (a few percentage points) and achieved under carefully controlled synthetic settings, which may not fully reflect real-world complexity. Moreover, the experiments primarily benchmark against older or relatively simple baselines, without comparison to stronger contemporary meta-learning or hybrid optimization approaches. Given the method’s high computational cost and limited novelty, the reported results, though positive, do not convincingly demonstrate a major performance breakthrough or justify the added algorithmic complexity.

**Questions:**

1. How does the proposed framework meaningfully advance beyond standard MAML or existing derivative-free meta-learning methods, and what concrete evidence supports the need to replace gradient-based approaches?

2. Given the high computational cost of two nested evolutionary loops, can the authors justify why such a design is necessary, and provide analysis demonstrating its scalability and efficiency in practical settings?

3. Since the algorithm appears to follow the OpenAI-ES formulation rather than true NES, and GPU offloading may be problem-dependent, how do the authors substantiate their claims of “natural gradient” estimation and general computational efficiency?

---

> ### Author Response · Authors · 2025-11-23
>
> We sincerely appreciate your valuable feedback. We have carefully considered your comments and provide our responses below.
>
> **W1.1-Limited novelty:** We appreciate this feedback. While NES and MAML are well-established, systematically integrate them as a functional algorithm is nontrivial. In particular, MAML is widely known to be difficult to implement [R1]. We resolve several key challenges by: 1) correctly formulating the bi-level problem of instance-level adaptation and NES gradients, 2) proposing two effective and novel gradient estimators for the inner loop and a correct first-order approximation for the outer loop, 3) achieving an efficient first-order outer-loop update to reduce overhead, and 4) a GPU parallelization scheme to enable practical training. We believe this is the first systematic integration of NES within a meta-learning framework for scheduling, and is a meaningful step toward practical and highly generalizable learning-based scheduling models.
>
> [R1] How to train your MAML. ICLR 2019
>
>
> **W1.2-Motivation for replacing gradient-based methods:** Thank you for the valuable comment. The advantage of the proposed framework over gradient-based methods is due to the following reasons.
>
> **First**, for the RL agent, solving FJSP and other combinatorial optimization problem is a task with *sparse reward*, since the reward signal is only generated at the end of each episode when a complete solution is constructed and the objective value is obtained. Gradient-based methods are known to struggle under sparse reward settings. To alleviate this issue, existing gradient-based scheduling methods such as (Zhang et al., 2020; Song et al., 2023; Wang et al., 2023) shape the reward to be denser by designing it as the makespan increment between two consecutive steps. However, the reward is still relatively sparse because not all actions lead to a makespan increment. We perform a quick test by running the DANIEL model on 100 instances from the 10×5 SD2 distribution, and found that on average only 15% timesteps in one episode have non-zero rewards. Therefore, the sparse reward issue largely remains. In contrast, our method directly work with sparse reward without the need of shaping, since the quality of the policy is simply evaluated based on complete solutions.
>
> To support the above analysis, we replace the reward of FOMAML to the sparse reward used in our model (i.e., a non-zero reward is obtained at the end of each episode). Results on the 10×5 SD2 dataset (Table R1) show that under this setting, the performance of gradient-based meta-learning further deteriorates, which validates our analysis.
>
> **Table R1.** Impact of Reward Sparsity on Gradient-Based Fine-Tuning
> ||Ours|FOMAML|FOMAML (sparse reward)|
> |:-:|:-:|:-:|:-:|
> |Greedy|369.2|394.4|400.8|
> |Sampling|347.7|356.0|361.2|
>
> **Second**, the fine-tuning process on an instance can be viewed as a search procedure for solving FJSP. Like other COPs, the solution space of FJSP is highly dicrete, non-smooth and non-convex. Gradient-based method would encounter substantial difficulties in this case, since the gradient could be highly unstable and the search could easily be trapped in local optimum. In contrast, our method could provide more robust gradient estimation for combinatorial search space, and have stronger global search ability hence is less prone to local optimum.
>
> To demonstrate this, we present the 10-step fine-tuning process of two representative instances with 13.6% and 25.8% non-zero reward in Tables R2 and R3, respectively. We can observe that on both instances, FOMAML easily gets trapped in local optimum, and the one with sparse reward performs even worse. EAS performs the best among gradient-based methods, since it directly use the objective function to update parameters and is less affected by the sparse reward issue, but it is still prone to local optimum. In contrast, the makespan of our method drops stably and is eventually lower than all gradient-based methods, showing a stronger search ability.
>
> **Table R2.** Instance-wies fine-tuning process (13.6% Non-zero Reward)
> ||step1|step2|step3|step4|step5|step6|step7|step8|step9|step10|
> |-|-|-|-|-|-|-|-|-|-|-|
> |Ours|460|452|466|485|452|437|439|421|421|421|
> |EAS|464|510|510|433|479|479|479|433|433|433|
> |FOMAML|463|463|463|463|479|479|479|479|479|479|
> |FOMAML (sparse reward)|463|463|463|480|480|480|480|491|491|480|
>
> **Table R3.** Instance-wies fine-tuning process (25.8% Non-zero Reward)
> ||step1|step2|step3|step4|step5|step6|step7|step8|step9|step10|
> |-|-|-|-|-|-|-|-|-|-|-|
> |Ours|454|422|388|394|386|373|368|373|366|368|
> |EAS|436|436|396|396|396|377|377|377|377|377|
> |FOMAML|439|439|439|439|402|402|402|402|394|394|
> |FOMAML (sparse reward)|439|439|439|439|439|436|436|439|439|439|

---

> > ### Author Response · Authors · 2025-11-23
> >
> > **W2&W3-Necessity of the nested evolutionary processes:** We fully understand your concern and would like to clarify this point as follows.
> >
> > **First**, the necessity of derivative-free meta-learning has been well justified by the strong empirical results in our paper and in our response above. Specifically, for instance-wise adapation, derivative-free method shows significant advantage over gradient-based methods, as shown in Table R2 and R3 above. This motivates us to develop a meta-learning framework that further exploits this advantage by providing a good initialization for more effective instance-wise adaptation. Importantly, it is a natural choice to make this framework fully derivative-free, because the inner loop is derivative-free and does not compute gradient, hence it is not efficient and not neccessary to use gradient-based methods in the outer loop.
> >
> > **Second**, we would like to respectfully clarify that our method does *not* involve two full ES loops. The inner loop is the only component that uses NES sampling. The outer loop employs the lightweight first-order approximation in Eq. (4), which only aggregates the inner-loop gradients and does not perform population sampling or second-order computations. As shown in Table R4,
> > the overhead of outer loop is very small. Thus, the concern about “two nested ES processes amplifying the cost” does not apply to our method.
> >
> > **Table R4.** Training Time Distribution Between Inner and Outer Loops on SD2 distributions
> > ||10×5|20×5|15×10|20×10|
> > |-|-|-|-|-|
> > |Inner loop|99.96%|99.97%|99.98%|99.98%|
> > |Outer loop|0.04%|0.03%|0.02%|0.02%|
> >
> > **Third**, we would like to respectfully clarify that our intention of using ES is to improve optimization performance, not to avoid computing second-order derivatives. Moreover, though backpropagation is not needed to compute exact second-order derivatives, applying ES in the outer loop to estimate them is costly as you mentioned. Therefore, we refer to the first-order approximation scheme in FOMAML, and develop a gradient-free version as stated in Eq. (4). So ES is *not* an alternative of other first-order or approximation approaches in gradient-based meta-learning methods. In fact, these gradient-based approaches can be utilized in our framework to derive other gradient-free counterparts, which is an interesting future work.
> >
> > **Finally**, we would like to respectfully clarify that while our method in the paper is trained for 200 epochs, its convergence is fast, achieving satisfactory results early in training. Here we list the performance of our method after 100 epochs on SD2 in Table R5, which shows that our model trained for 100 epochs performs nearly identically to the one trained for 200 epochs, while the training time is only half as in Table R6. So in practice, early stopping is possible to obtain good performance.
> >
> > **Table R5.** Comparison of solution quality
> > ||10×5|20×5|15×10|20×10|30×10|40×10|
> > |-|-|-|-|-|-|-|
> > |FOMAML|394.4|646.6|557.1|583.8|757.6|949.3|
> > |Ours(100 Epochs)|369.2|625.1|531.8|561.7|732.8|920.5|
> > |Ours(200 Epochs)|369.2|624.8|531.4|559.9|732.6|920.1|
> >
> > **Table R6.** Comparison of training time
> > ||10×5|20×5|15×10|20×10|
> > |-|-|-|-|-|
> > |FOMAML|1.5h|2.9h|3.7h|5.5h|
> > |Ours(100 Epochs)|1h|2.3h|3.6h|5.5h|
> > |Ours(200 Epochs)|2.1h|4.5h|7.2h|11.2h|
> >
> > To summarize, our method is based on reasonable and natural design choices, instead of arbitrary or unnecessary complexity. We will clarify these points in the revised paper.
> >
> > **W4-Misuse of natural gradient:** We sincerely appreciate the reviewer's careful reading and this important terminology clarification. We would like to respectfully address your concern as follows. Indeed, our method follows OpenAI-ES, which belongs to NES as mentioned in Section 2 in Salimans et al. (2017). It implements the search distribution as an isotropic Gaussian with fixed covariance. In this specific but widely used formulation, Section 3.3 ("Rotationally Symmetric Distributions") of Wierstra et al. (2014) shows that the Fisher information matrix becomes a scalar multiple of the identity. Under this condition, the natural gradient direction coincides with the standard log-likelihood gradient estimator up to a constant factor. Consequently, though we do not compute or invert the full Fisher information matrix, our implementation is a variant of NES and aligns with the natural gradient direction in the canonical NES setting up to a constant factor. In light of the reviewer's concern, we realize that this implicit assumption may not have been stated clearly enough in the paper. We have carefuly revised our paper, especially Section 3, to minimize possible confusions and misleading statements.

---

> > > ### Author Response · Authors · 2025-11-23
> > >
> > > **W5-Feasibility and generality of GPU parallelization:** Thank you for this insightful comment. We fully agree that the degree to which the environment itself can be executed on GPU varies greatly across scenarios, and full GPU offloading is not always feasible. Here we would like to clarify that the efficiency gain of our approach primarily stems from the parallelization of the population network, which is GPU-friendly and generally applicable. As shown in Table R7 below, this mechanism accounts for more than 95% of the training time reduction with respect to full CPU training (90% of the 120h is on network inference). Therefore, even if the environment remains purely CPU-based, the method still benefits substantially from GPU acceleration. Full GPU offloading of the environment yields an additional incremental gain, but it is not required for our method to be efficient.
> > >
> > > **Table R7.** Training Time Comparison on 10×5 SD2 distribution
> > > | Method | Training Time |
> > > |---|---|
> > > | Full CPU (with Ray-based multithreading) | 120h |
> > > | Network Parallelized Only | 5h |
> > > | Network and Environment Fully Parallelized | 2h |
> > >
> > > We will carefully revise the manuscript to clarify that our efficiency claims mainly refer to model-side parallelization, which generalizes to different environments, while full environment GPU execution is optional and problem-dependent.
> > >
> > >
> > > **W6-Modest improvements, synthetic data, and additinoal complexity:** We fully understand your concern and would like to address it as follows.
> > >
> > > _**On the magnitude of improvements and choice of baselines.**_ We acknowledge the reviewer’s point that the improvements over existing baselines such as AS and EAS are incremental rather than dramatic. However, we would like to respectfully clarify that in the context of FJSP and related COP settings, even small improvements in the objective can lead to substantial economic gain or cost reduction in reality. Our baselines DANIEL/SPN represent state-of-the-art research in learning-driven scheduling, and AS/EAS are the strongest instance-wise adapation methods to the best of our knowledge. Our consistent improvements over these methods, even if numerically modest, are therefore meaningful. In particular, we successfully demonstrated that derivative-freemeta-learning significantly accelerates and stabilizes instance-wise adaptation, and scales to very large instances. These aspects represent practical and non-trivial advantages that conventional baselines do not possess.
> > >
> > > _**On the synthetic data.**_ We would like to note that the synthetic distributions used in our experiments strictly follow existing literature, which are specifically designed to capture the complexity of real-world (flexible) job shop environments. Importantly, in addition to the synthetic datasets, we have also validated the effectiveness of our method on publicly JSP/FJSP instances that are well-acknowledged as standard benchmarks, as shown in Tables 2 and 10 of the paper. Moreover, the large-scale instances in Table 8 (up to 100×20) demonstrate that our method remains effective and stable even when scaling far beyond the distribution seen during meta-training.
> > >
> > > _**On the additional complexity.**_ We acknowledge that just like standard meta-learning methods, our method brings additional complexity comparing to the non-meta-learning methods AS/EAS. However, we believe the additional complexity is reasonable. First, the learned initialization indeed leads to better fine-tuning performance, with the same or even shorter runtime, as already been demonstrated in our paper. Second, as explained in Table R6 above, the offline training cost is reasonable. With the same or even shorter training time, our meta-model exhibits much better performance than FOMAML, showing the necessity of a gradient-free meta-learning framework.

---

> > > > ### Author Response · Authors · 2025-11-23
> > > >
> > > > **Q1-Advance beyond standard MAML or existing derivative-free meta-learning methods:** We sincerely appreciate your valuable feedback. The advancement over standard MAML has been explained in our response to W1.2 above. The advancement over existing derivative-free meta-learning methods is explained as follows.
> > > >
> > > > **First**, existing derivative-free optimization methods rely on CPU parallization, which hinders their applications in complex meta-learning scenarios. In contrast, by leveraging the population-based nature and high parallelism of Evolution Strategies, we developed a highly parallelized GPU population network. As shown in Table 4, our approach achieves an order of magnitude improvement in population forward propagation efficiency. This advancement makes it feasible to apply DFO methods in instance-level meta-learning.
> > > >
> > > > **Second**, existing derivative-free optimization methods use a single policy rollout to evaluate fitness, which tend to be unstable in complex scheduling problems. To stabilize DFO training within the instance-level inner loop, we propose two Monte Carlo (MC) estimation strategies: MC averaging and MC best-sample. We run a quick test on the SD2 dataset to compared our MC averaging with the standard single-sample update methods under the greedy mode, and the results in Table R8 further demonstrat the effectiveness of our design.
> > > >
> > > > **Table R8.** Comparison of inner-loop update methods on SD2 under the greedy mode
> > > > ||10×5|20×5|15×10|20×10|
> > > > |-|-|-|-|-|
> > > > |Standard single-sample|378.6|635.7|543.6|568.0|
> > > > |MC averaging|369.2|624.8|531.4|559.9|
> > > >
> > > >
> > > >  **Q2-Necessity of the nested evolutionary processes:** Please see the above response to **W2&W3**.
> > > >
> > > >  **Q3.1-Misuse of natural gradient:** Please see the above response to **W4**.
> > > >
> > > >  **Q3.2-Feasibility and generality of GPU parallelization:** Please see the above response to **W5**.

---

> > > > > ### Comment · Reviewer_kVaC · 2025-11-23
> > > > >
> > > > > Thank you for the detailed rebuttal and additional experimental results. The clarifications regarding sparse reward handling, GPU-side network parallelization, and the structure of the inner/outer optimization loops are helpful, and they address several issues in my original comments.
> > > > >
> > > > > However, the rebuttal does not fully resolve some core concerns I had. Specifically, the question of conceptual novelty remains largely unanswered: the method still appears to be a combination of MAML-style meta-learning with ES-based adaptation, without a compelling theoretical argument or comparative study demonstrating that this integration yields fundamentally new insights or capabilities beyond prior derivative-free meta-learning work. The justification for adopting ES in particular remains empirical and limited to selected comparisons, without evaluation against other first-order or gradient-free meta-learning alternatives that could avoid second-order costs with substantially lower sample complexity.
> > > > >
> > > > > The terminology clarification regarding “natural gradient” relies on restrictive assumptions about isotropic Gaussian distributions and equivalence up to scaling, which does not fully address the earlier concern that the paper presents this aspect as a methodological contribution. The rebuttal effectively confirms that the approach follows the OpenAI-ES update rule rather than performing true NES-style natural gradient computation, which weakens the conceptual framing.
> > > > >
> > > > > Furthermore, although GPU batching reduces wall-clock time, the fundamental sample inefficiency of large-population ES remains, and the provided results do not convincingly demonstrate scalability or practicality beyond controlled synthetic and benchmark settings.

---

> > > > > > ### Comment · Reviewer_kVaC · 2025-11-23
> > > > > >
> > > > > > Regarding the theoretical soundness of the proposed framework, I remain unconvinced that the method is optimizing the stated bi-level meta-learning objective in a principled way. Although the rebuttal provides additional empirical evidence that the ES-based inner loop can improve performance under sparse rewards, it still does not demonstrate that the outer-loop update constitutes a correct or consistent meta-gradient. In particular, the outer loop does not differentiate through the adaptation process, but instead seems to aggregates ES-derived fitness estimates. Therefore, it may effectively optimize a smoothed surrogate objective rather than the meta-objective formulated in Eq. (1). Consequently, it remains unclear whether the meta-update truly enhances the model’s ability to adapt, or whether it provides any guarantees of convergence or generalization across instances. Overall, while the heuristic may work empirically, the overall approach (especially the outer loop) may remain theoretically underdeveloped and the conceptual grounding may be overstated.

---

> > > > > > ### Author Response · Authors · 2025-11-27
> > > > > >
> > > > > > We sincerely appreciate reviewer's careful reconsideration and are glad that our responses address part of your concerns. Below we further address the remaining issues raised by the reviewer.
> > > > > >
> > > > > > **Limited novelty:** Thank you for your important comment. While our method indeed combines meta-learning with ES adaptation, we would like to emphasize that the novelty does not lie in introducing a new optimization primitive, but in addressing a specific and previously unsolved challenge in learning-based combinatorial optimization. The specific justifications are as follows.
> > > > > >
> > > > > > - First, to the best of our knowledge, our method is the *first* effective derivative-free meta-learning method that works for scheduling, a complicated COP task. As mentioned in the related work, existing instance-wise adaptation methods are all gradient-based, which struggles in optimization problems with highly discontinuous search space and irregular objective landscape as demonstrated in our previous response. It is even counterproductive to introduce gradient-based meta-learning in this case. Our method is conceptually different from existing works, and we show that derivative-free meta-learning can meaningfully optimize initializations for instances-wise adaptation, leading to significantly better and more stable performance and even scales to very large-scale instances beyond the reach of gradient-based methods.
> > > > > >
> > > > > > - Second, we would like to respectfully emphasize that it is not straightforward to make the combination of meta-learning and ES work. Three non-trivial technical challenges must be addressed: 1) the single-sample evaluation in standard ES suffers from high variance for the inner loop gradient estimation; 2) nesting a full ES in the outer loop is numerically unstable and costly; and 3) existing ES implementations rely on CPU parallelization, making the training time a major bottleneck for practical use. We resolve these challenges by: 1) proposing two multi-sample MC estimators that significantly improves the fine-tuning performance; 2) design a lightweight outer loop update rule inspired by FOMAML; and 3) a population-level parallel fitness evaluation method that fully exploit GPU to significantly improve training efficiency. All these parts contribute to the final gains as validated in our ablation study.
> > > > > >
> > > > > > In short, while the building blocks are known, our work resolves the key obstacles that previously prevented derivative-free meta-learning from being effective in complex scheduling environments. We believe this is a meaningful step toward advancing learning-based scheduling.
> > > > > >
> > > > > >
> > > > > > **Justification for adopting ES:** We appreciate this insightful feedback and would like to provide further clarification. Our primary motivation for employing Evolution Strategies (ES) is to enhance the performance of instance-level fine-tuning. The avoidance of second-order derivatives is a natural consequence of our design choice, as we adopt a first-order outer-loop update mechanism, rather than being the primary reason for using ES. The first-order approximation in our method directly follows the gradient-based FOMAML framework.
> > > > > >
> > > > > > To assess the performance of alternative first-order strategies, we include a comparison on the 10x5 SD2 dataset with another seminal first-order approximation approach, Reptile [R1]. The results presented in Table R1 demonstrate that, because Reptile does not rely on gradients for updating the global model parameters during the meta-update step, it improves the performance of gradient-based meta-learning methods. However, our gradient-free version of the framework still significantly outperforms the gradient-based Reptile, underscoring the effectiveness of our gradient-free approach.
> > > > > >
> > > > > > **Table R1.** Performance Comparison of First-Order Approximation Methods (FOMAML vs. Reptile) on 10×5 SD2
> > > > > > ||Gradient-based Meta-Learning|Gradient-free Meta-Learning (ours)|
> > > > > > |-|-|-|
> > > > > > |FOMAML |394.4|369.2|
> > > > > > |Reptile|391.1|371.5|
> > > > > >
> > > > > > [R1] On first-order meta-learning algorithms. arXiv preprint arXiv:1803.02999.

---

> > > > > > > ### Author Response · Authors · 2025-11-27
> > > > > > >
> > > > > > > **Application of the natural gradient terminology:** We sincerely respect your rigorous approach to the terminology in the paper. We would like to respectfully address your further concerns as follows.
> > > > > > >
> > > > > > > First, we acknowledge that our method does not compute the *true* natural gradient. As mentioned in our previous response to W4, OpenAI-ES is a variant of NES as described in Section 2 of Salimans et al. (2017). This can also be supported by Wierstra et al. (2014), the seminal paper of NES, where Section 3.3 shows that when the covariance is fixed and isotropic, the natural gradient *reduces* to this form, avoiding the need to estimate/invert a full covariance matrix. Consequently, the ES gradient computed here is a variant (or approximation) of natural gradient.
> > > > > > >
> > > > > > > Second, we did not intend to position natural gradient computation as a methodological contribution of the paper. Our aim was only to situate our approach within the broader NES family of derivative-free optimization methods, in which the OpenAI-ES estimator is frequently  described as an instance of natural-gradient evolution strategies under isotropic Gaussian parameterization. In light of the reviewer's concern, we have revised the description of the terminology to ensure the highest level of precision in the paper.
> > > > > > >
> > > > > > > Finally, we would like to gently point out that the terminology issue does not affect the technical core of the paper. Our conceptual framing, as reflected in the title, lies in the effective integration of derivative-free methods with instance-level meta-learning, rather than the specific choice of a derivative-free method. As such, the overall framework remains unaffected, and combining it with other derivative-free methods will be a key direction for future work.
> > > > > > >
> > > > > > > We hope this clarification resolves the concern and sincerely appreciate the reviewer’s detailed attention to conceptual accuracy.
> > > > > > >
> > > > > > >
> > > > > > > **Large-population inefficiency:** Thank you for this valuable feedback. We fully understand the concerns regarding the practical efficiency. We would like to respectfully address your concerns as follows.
> > > > > > >
> > > > > > > First, while ES indeed requires a larger population per update, it is a valuable alternative to gradient-based methods due to the advantages of indifference to the distribution of rewards (sparse or dense), no need for backpropagating gradients, and tolerance of potentially arbitrarily long time horizons (Salimans et al. 2017). In our case, gradient-based methods suffer from relatively poor ability in instance-wise adapation and meta-model training, and fails in scaling to very large-scale problems. Our gradient-free methods succesfully overcome these limitations, showing a strong empirical performance.
> > > > > > >
> > > > > > > Second, the practical efficiency of our method is competitive, and simple trade-offs can further reduce its cost. As shown in Table R2 and R3 in our previous response, early stopping at 100 epochs yields nearly identical performance to 200 epochs while reducing training time to a level comparable to (or even lower than) gradient-based meta-learning baselines. Here we supplement new experiments of using a smaller population of size 50 on the SD2 dataset, as shown in Table R2 and R3. Results reveal only a slight decrease in performance, which still outperforms all baseline methods comprehensively, while also significantly reducing memory usage, as detailed in Table R3.
> > > > > > >
> > > > > > > **Table R2.** Comparison of solution quality
> > > > > > > ||10×5|20×5|15×10|20×10|
> > > > > > > |-|-|-|-|-|
> > > > > > > |Gradient-based FOMAML|394.4|646.6|557.1|583.8|
> > > > > > > |EAS|380.5|640.8|547.7|569.1|
> > > > > > > |Ours-50 Population|374.2|627.3|533.1|562.2|
> > > > > > > |Ours-100 Population|369.2|624.8|531.4|559.9|
> > > > > > >
> > > > > > > **Table R3.** Comparison of training cost
> > > > > > > ||10×5|20×5|15×10|20×10|
> > > > > > > |-|-|-|-|-|
> > > > > > > |Gradient-based FOMAML|0.9G|1G|1G|1.2G|
> > > > > > > |Ours-50 Population|0.9G|1.1G|1.2G|1.3G|
> > > > > > > |Ours-100 Population|1.3G|1.6G|2.0G|2.2G|
> > > > > > >
> > > > > > > Finally, we would like to emphasize that the goal of the proposed framework is not to reduce the theoretical sample complexity of ES, but to make derivative-free adaptation practical and competitive for difficult scheduling problems through a combination of meta-learning and GPU-parallel evaluation. The evidence from large-scale experiments, ablation studies, and fine-tuning curves collectively supports this claim.
> > > > > > >
> > > > > > > We hope this clarification resolves the concern and sincerely appreciate the reviewer’s thoughtful perspective on scalability.

---

> > > > > > > > ### Author Response · Authors · 2025-11-27
> > > > > > > >
> > > > > > > > **Theoretical soundness of the proposed framework:** We sincerely appreciate the reviewer's valuable feedback. We would like to respectfully clarify that not differentiating through the adaptation process is a defining property of first-order meta-learning methods. Both FOMAML and Reptile intentionally avoid backpropagating through the inner-loop update and therefore do not compute the exact bi-level meta-gradient. Our Eq. (4) follows the principle of FOMAML (see Eq.(8) in [R2]), replacing gradient-based inner updates with ES-based estimation. Consequently, our method is consistent with the established theoretical framework of gradient-based first-order meta-learners. Table 3 in our paper has already shown that our meta-update truly enhances the model's ability to adapt, where ES means training the DANIEL model without meta-gradient. We further add the detailed fine-tuning process of both methods on 10x5 SD2 instances in Table R4. Both Table 3 and Table R4 clearly demonstrate that our method’s meta-gradient in the outer loop results in faster convergence during adaptation and better solution quality.
> > > > > > > >
> > > > > > > > **Table R4.** Comparison of Fine-Tuning Performance
> > > > > > > > ||step1|step2|step3|step4|step5|step6|step7|step8|step9|step10|
> > > > > > > > |-|-|-|-|-|-|-|-|-|-|-|
> > > > > > > > |Ours|416.8|409.9|398.9|397.1|391.3|389.3|386.7|386.9|386.0|382.7|
> > > > > > > > |ES|419.2|414.5|406.9|401.8|401.3|400.1|398.1|399.2|396.6|394.8|
> > > > > > > >
> > > > > > > > [R2] On the Convergence Theory of Gradient-Based Model-Agnostic Meta-Learning Algorithms. AISTATS 2020.

---

### Official Review · Reviewer_2y5p · 2025-10-25

**Soundness:** 3
**Presentation:** 4
**Contribution:** 3
**Rating:** 6
**Confidence:** 4

**Summary:**

This paper addresses a key limitation of learning-based approaches for NP-hard scheduling problems: models are often trained to optimize average performance, rather than the solution quality for each specific instance. Existing instance-level adaptation methods are typically used only at test time and rely on gradient-based optimization, which can be ineffective in combinatorial search spaces.

Inspired by MAML, the authors propose an instance-level meta-learning framework to learn a model initialization that can quickly adapt to new instances. Its main novelty lies in the use of a *fully derivative-free optimization (DFO)* method, specifically Natural Evolution Strategies (NES), for both the inner-loop adaptation and the outer-loop meta-optimization. This avoids the need for complex gradient calculations and is better suited for instance-level search tasks, which can trap gradient-based methods in local optima.

To stabilize the DFO process, the paper introduces two Monte Carlo estimation strategies (MC averaging and MC best-sample). Furthermore, they develop an efficient GPU-based parallel framework to manage the computational cost of the population-based NES.

The effectiveness of the method is demonstrated through extensive experiments on Flexible Job Shop Scheduling (FJSP) and Job Shop Scheduling (JSP), applying it to two base models: Reinforcement Learning (DANIEL) and Self-supervised (SPN). The results show consistent and significant improvements over the base models and sota adaptation baselines, such as Active Search (AS) and Efficient Active Search (EAS).

**Strengths:**

1. The paper addresses a critical and practical problem in Neural Combinatorial Optimization (NCO).
2. The core contribution—a fully derivative-free meta-learning framework for instance-level NCO. While MAML and NES are existing methods, their combination to solve the specific limitations of gradient-based instance adaptation in scheduling is a sufficient methodological contribution.
3. The comprehensive experiments are a strength.
   - The method is tested on two different scheduling problems (FJSP, JSP), two different learning paradigms (RL and self-supervised), multiple synthetic datasets (SD1, SD2), and standard public benchmarks (mk, la, Taillard).
   - Comparison against strong and relevant baselines, including exact solvers (OR-Tools), strong heuristics (MWKR), base models (DANIEL, SPN), and state-of-the-art gradient-based adaptation methods (AS, EAS).
   - Sufficient ablation studies: DFO vs. Gradient Adaptation, DFO-Meta vs. DFO-only, DFO-Meta vs. Gradient-Meta, the proposed MC estimators, and GPU parallelization.
4. The method performs well on both the RL-based DANIEL and the self-supervised SPN. Furthermore, the scalability experiments on larger $50 \times 20$ and $100 \times 20$ instances (Table 8) are impressive, highlighting a benefit of the DFO approach: much higher memory efficiency than gradient-based adaptation (AS/EAS).
5. The motivation is clear, the logical flow is sound, and it effectively uses figures (Figs 1, 2) and algorithms (Algs 2, 3) to explain the complex framework.

**Weaknesses:**

1. **Training Cost:** The main drawback is the computational cost of DFO, which the authors acknowledge. Appendix C confirms the method requires approximately 1.5-2x the training time and GPU memory of the gradient-based FOMAML baseline. While the authors argue this is a reasonable trade-off for offline training, the high cost of population-based methods remains a significant barrier. A population size of $\mu=100$ and 200 epochs represent a substantial computational investment.
2. **Hyperparameter Sensitivity:** NES and other DFO methods are often highly sensitive to hyperparameters (e.g., population size $\mu$, noise standard deviation $\sigma$, step sizes $\alpha, \beta$). The paper states these were tuned on the smallest instance size and fixed, which is a reasonable protocol. However, the paper lacks a sensitivity analysis. Given that DFO methods can be "notoriously" difficult to tune, including a brief analysis (e.g., in the appendix) showing how performance varies with different $\mu$ or $\sigma$ values would significantly enhance the paper's practical value and demonstrate the method's robustness. The sensitivity of DFO methods like NES to hyperparameters such as population size $\mu$, noise standard deviation $\sigma$, and learning rates $\alpha, \beta$ is well-known. The paper fails to provide any sensitivity analysis for these critical hyperparameters. This calls the reproducibility and robustness of the experimental results into serious question. We cannot judge whether the current SOTA results are the product of immense hyperparameter-babysitting or if the method is genuinely robust.
3. The paper's core contribution is an effective combination of two well-known techniques—MAML (meta-learning) and NES (derivative-free optimization)—applied to a known problem (instance-level adaptation). Neither meta-learning for NCO nor instance-level adaptation (as shown by AS/EAS) are entirely new concepts.
4. The choice of NES (a DFO method) as the core optimization strategy might be a "brute-force" choice. It essentially uses massive computational resources and sample complexity (population-based search) to bypass the (researchable) challenges faced by gradient-based adaptation. One of the key comparisons in the paper (Table 5) shows that the authors' proposed DFO method offers a relatively weak performance improvement over standard gradient meta-learning (FOMAML) (e.g., on $20 \times 10$ SD2, greedy mode only improves from 21.22% to 23.21%, and sampling mode only from 14.72% to 19.36%). To achieve this slight improvement, Appendix C (Table 9) shows that the authors' method requires **1.5 to 2 times** the training time and GPU memory. Exchanging 2x the computational resources for a 2-4% performance gain makes the method's adoption in other domains difficult.
5. DFO not needing to store gradients at inference time, thus having low memory usage, is its inherent characteristic, but its training-stage cost (Table 9) is large (e.g., $11.2$ hours for $20 \times 10$).

**Questions:**

1. **Training Time vs. Baselines:** Appendix C provides an excellent comparison of training costs against FOMAML. However, how does the total meta-training time (e.g., 11.2 hours for $20 \times 10$) compare to the *original pre-training time* of the DANIEL baseline? Understanding the *total* training budget (pre-training for DANIEL vs. meta-training for the proposed method) would provide a clearer picture of the overall computational trade-off. Training a model that requires 11 hours ($20 \times 10$) suggests that addressing the conflict between "inference-time memory advantage" and "training-time catastrophic cost" might be a necessary component.
2. **Rationale for MC Estimators in Ablation:** In Section 5.2, the paper states "MC averaging" is used for greedy mode and "MC best-sample" for sampling mode. This is intuitive. However, in the ablation study (Figure 3), all estimators (including MC averaging) are evaluated in *sampling mode*. Can you clarify this? Was MC averaging also evaluated in greedy mode? Was the MC best-sample strategy *also* shown to be suboptimal for greedy mode, or is MC averaging indeed the best choice for greedy mode?
3. **"Perturbation-on-Perturbation" Variance:** The paper makes an interesting point in 4.1.2 that the "perturbation-on-perturbation" mechanism increases the variance of the natural gradient estimate. Does this imply that the "NES" baseline in Table 3 (i.e., DFO fine-tuning *without* meta-learning) is more stable or converges faster (per-instance) during its test-time adaptation, even if its final solution quality is worse?
4. **Sensitivity to Population Size:** The population size $\mu=100$ seems to be a critical hyperparameter. What was the rationale for choosing this specific value? How much does performance degrade if a smaller, computationally "cheaper" population is used (e.g., $\mu=20$ or $\mu=50$)? A sensitivity analysis on key NES hyperparameters (especially population size $\mu$ and noise $\sigma$) may be needed. This is a core issue for DFO methods. Without this analysis, reviewers cannot assess whether your method is robust or requires extreme tuning to achieve the claimed performance.
5. Given that the authors' method (Appendix C) requires up to 2x the training overhead, while the performance gain over standard gradient meta-learning FOMAML (Table 5) is small, strong evidence may be needed to prove that this complex, computationally expensive DFO framework truly offers a substantial advantage over simple FOMAML. An elaboration on this would strengthen the paper.
6. **Justification for MC Best-Sample:** Using MC best-sample (Eq. 3) to estimate fitness in the inner loop seems to "overfit" to the "sampling mode" during the meta-training phase. Does this estimator harm the model's performance in "greedy mode"? Did the authors compare whether the greedy mode performance would be better if MC averaging was used uniformly?

---

> ### Author Response · Authors · 2025-11-23
>
> We greatly appreciate your valuable feedback, and have carefully addressed your concerns as follows.
>
> **W1&W4&W5-Training cost vs FOMAML and DANIEL:** Thank you for the important comment. We would like to address your concern by providing further clarification on the matter.
>
> **On one hand**, the low training cost of FOMAML is somewhat misleading. Combining the results in Table 5 and Table 1, we can see that the majority of FOMAML's performance is inferior to that of EAS and AS which do not employ meta-learning. In other words, its fast training is actually of no significant meaning here since it is counterproductive to perform gradient-based meta-learning.
>
> **On the other hand**, while our method in the paper is trained for 200 epochs, its convergence is fast, achieving satisfactory results early in training. Here we list the performance of our method after 100 epochs on SD2 in Table R1, which shows that our model trained for 100 epochs performs nearly identically to the one trained for 200 epochs, while the training time is only half as in Table R2. So in practice, early stopping is possible to obtain good performance.
>
> **Table R1.** Comparison of solution quality
> ||10×5|20×5|15×10|20×10|30×10|40×10|
> |-|-|-|-|-|-|-|
> |DANIEL|408.4|671.0|591.2|610.1|774.6|962.6|
> |FOMAML|394.4|646.6|557.1|583.8|757.6|949.3|
> |Ours(100 Epochs)|369.2|625.1|531.8|561.7|732.8|920.5|
> |Ours(200 Epochs)|369.2|624.8|531.4|559.9|732.6|920.1|
>
> As shown in Table R2, the training time of both our method and FOMAML is longer than that of DANIEL. This is expected since in each iteration, the inner loop performs multiple updates for each instance, increasing the training time. However, this additional training time results in significantly better instance adaptation and generalization performance, especially on large-scale problems (sizes 50×20 and 100×20).
>
> **Table R2.** Training Time Comparison
> ||10×5|20×5|15×10|20×10|
> |-|-|-|-|-|
> |DANIEL |0.2h|0.4h|0.7h|1.0h|
> |FOMAML |1.5h|2.9h|3.7h|5.5h|
> |Ours(100 Epochs)|1h|2.3h|3.6h|5.5h|
> |Ours(200 Epochs)|2.1h|4.5h|7.2h|11.2h|
>
>
> **W2-Hyperparameter sensitivity:** Thank you for the valuable comment. We examine the impact of four key hyperparameters on NES : population size $\mu$, standard deviation $\sigma$ of the search distribution, number of inner-loop steps $K$, and update step sizes $\alpha$ and $\beta$. Experiments are conducted on the 10×5 SD2 dataset (used for hyperparameter tuning). As shown in Table R3, a larger population leads to better performance but significantly increases training time. We set $\mu=100$ to achieve a reasonable tradeoff between solution quality and training cost. Tables R4, R5, and R6 show that the impacts of $\sigma$, $K$, $\alpha$, and $\beta$ are relatively small, with $\sigma = 0.2$, $K = 3$, and $\alpha = \beta = 0.05$ yielding the best performance.
>
> **Table R3.** Impact of population size $\mu$
> ||$\mu=20$|$\mu=50$|$\mu=100$|$\mu=300$|
> |-|-|-|-|-|
> |Performance|377.5|374.2|369.2|365.5|
> |Training time|1.7h|1.8h|2.1h|3.3h|
>
> **Table R4.** Impact of standard deviation $\sigma$
> ||$\sigma=0.05$|$\sigma=0.1$|$\sigma=0.15$|$\sigma=0.2$|$\sigma=0.25$|$\sigma=0.3$|
> |-|-|-|-|-|-|-|
> |Performance|375.6|372.3|372.6|369.2|374.2|375.4|
> |Training time|2.1h|2.1h|2.1h|2.1h|2.1h|2.1h|
>
> **Table R5.** Impact of inner-loop steps $K$
> ||$K=1$|$K=2$|$K=3$|$K=4$|$K=5$|
> |-|-|-|-|-|-|
> |Performance|372.9|369.7|369.2|370.6|370.2|
> |Training time|1.2h|1.7h|2.1h|2.6h|3.1h|
>
> **Table R6.** Impact of step sizes $\alpha$ and $\beta$
> || $\alpha = \beta = 0.03$| $\alpha = \beta = 0.04$| $\alpha = \beta = 0.05$| $\alpha = \beta = 0.06$| $\alpha = \beta = 0.07$|
> |:-:|:-:|:-:|:-:|:-:|:-:|
> |Performance|374.3|370.5|369.2|372.5|370.9|
> |Training time|2.1h|2.1h|2.1h|2.1h|2.1h|
>
>
> **W3-Combination of two well-known techniques:** Thanks for the valuable comment. While NES and MAML are well-established, we would like to clarify that our work is not a simple combination but a systematic adaptation and integration of the two methods for instance-level adaptation. This is challenging and nontrivial, and we successfully addressed several key challenges:
>
> - A sound and novel bi-level optimization formulation for instance-wise adaptation with NES gradients;
>
> - The design of two effective inner-loop gradient estimators that address the high-variance of derivative-free optimization;
>
> - A correct and efficient first-order outer-loop update to reduce overhead;
>
> - A GPU-parallelized training scheme that significantly improves the scalability and practicality.
>
> We believe this is the first systematic integration of NES within a meta-learning framework for scheduling, and is a meaningful step toward practical and highly generalizable learning-based scheduling models.

---

> > ### Author Response · Authors · 2025-11-23
> >
> > **Q1-Training cost vs DANIEL:** Please see the above response to **W1&W4&W5**.
> >
> > **Q2&Q6-MC estimators in ablation: mode evaluation:** Thank you for this insightful comment. In the ablation study in Figure 3, we aim to demonstrate that the MC best-sample performs better than MC averaging in the sampling mode, since this mode leads to the best optimization performance. Following your suggestion, here we conduct further comparison between MC best-sample and MC averaging in the greedy mode. Results in Table R7 indicates that MC averaging slightly outperforms MC best-sample in the greedy mode, confirming the reviewer’s intuition: each estimator indeed tends to favor the mode it is conceptually aligned with.
> >
> > Importantly, the difference under greedy mode is small, while MC best-sample offers clear benefits in sampling mode, which is the stronger inference mode in our framework. Thus, MC best-sample is a good default choice in practice, while MC averaging can be used if greedy performance is prioritized under time-critical applications.
> >
> > **Table R7.** Comparison of MC averaging and MC best-sample in the greedy mode
> > ||10×5|20×5|15×10|20×10|30×10|40×10|
> > |-|-|-|-|-|-|-|
> > |MC Averaging|369.2|624.8|531.4|559.9|732.6|920.1|
> > |MC Best-Sample|369.7|626.3|533.5|560.3|732.9|921.3|
> >
> >
> > **Q3-Comparison of meta-learning and NES adaptation:** Thank you for this insightful comment. We would like to clarify that the "perturbation-on-perturbation" mechanism affects the meta-training phase, where the natural gradient estimate may indeed have higher variance if computed exactly. This is why our method adopts the first-order approximation in the outer loop, which keeps the meta-gradient stable and reduces computational overhead.
> >
> > Importantly, during test-time adaptation, our meta-model and the NES baseline both use exactly the same first-order NES update rule, and there is no additional higher-order perturbation. Therefore, the increased variance during meta-training does not imply that the meta-model would be less stable than NES during instance-wise adaptation.
> >
> > Here we follow your suggestion and compare the fine-tuning performance of NES and our meta-model on 10x5 instances from SD2, as shown in Table R8. It has been demonstrated that the meta-model converges faster during adaptation and yields better solution quality. This could be attributed to the fact that our meta-learning method explicitly simulates and optimizes the fine-tuning process, hence provides a better initialization that leads to better instance-wise adapation.
> >
> > **Table R8.** Comparison of Fine-Tuning Performance
> > ||step1|step2|step3|step4|step5|step6|step7|step8|step9|step10|
> > |-|-|-|-|-|-|-|-|-|-|-|
> > |Ours|416.8|409.9|398.9|397.1|391.3|389.3|386.7|386.9|386.0|382.7|
> > |NES|419.2|414.5|406.9|401.8|401.3|400.1|398.1|399.2|396.6|394.8|
> >
> >
> > **Q4-Hyperparameter sensitivity:** Please see the above response to **W2**.
> >
> >
> > **Q5-Concerns on performance gain and computational overhead:** Please see the above response to **W1&W4&W5**.

---

### Official Review · Reviewer_3mss · 2025-10-29

**Soundness:** 3
**Presentation:** 3
**Contribution:** 3
**Rating:** 6
**Confidence:** 3

**Summary:**

The paper addresses the problem that existing DRL approaches for job-shop and flexible job-shop scheduling optimize the average performance across training instances, which often leads to poor performance on specific individual instances. Existing test-time adaptation methods improve individual instances through fine-tuning, but they operate only at inference time, do not transfer adaptation knowledge across instances, and rely on gradient-based optimization, which can be ineffective in combinatorial search settings.

The authors propose a derivative-free instance-wise meta-learning framework. Instead of learning a single general policy, the method meta-learns an initialization of the policy parameters that is explicitly optimized to be adapted efficiently to each new scheduling instance. The approach uses NES to perform both inner-loop adaptation and outer-loop meta-updates, avoiding gradients entirely. Additionally, the paper introduces a GPU-parallelized population evaluation mechanism, enabling efficient large-scale derivative-free optimization.

Experiments on FJSSP (and additionally JSSP with a non-RL model) show that the meta-learned initialization enables faster and better per-instance adaptation than prior approaches, and generalizes to larger unseen scheduling instances.

**Strengths:**

- The authors explicitly shift the learning objective from maximizing average policy performance to optimizing instance-wise adaptability, which departs from the dominant paradigm in DRL-based scheduling. In addition, the paper contributes a system-level innovation, parallelizing population evaluations on GPUs. Hence, the paper offers a novel combination of ideas, which to my knowledge has not been applied in NCO. While individual components (MAML, NES, DRL scheduling) exist, the integration and the shift in optimization objective represent a meaningful contribution.

- The authors claim that their meta-learned initialization enables more effective per-instance adaptation than gradient-based approaches. The experiments include multiple baselines (heuristic, OR-Tools, DRL, AS, EAS) and include ablation studies that isolate the effect of meta-learning and derivative-free optimization independently.

- The experimental setup is thorough and follows best practice: standard benchmark datasets and larger unseen instances are evaluated. Further, hyperparameters and architectures are reported in sufficient detail for reproducibility. The ablations and generalization tests strengthen the claim that improvements arise from the proposed components rather than from tuning or dataset-specific effects.

**Weaknesses:**

- The paper combines multiple complex components at once (meta-learning, NES, scheduling, PPO), but the authors do not clearly modularize the explanation. Heavy notation is introduced early, creating a high cognitive load before the conceptual flow is fully established. Although figures exist, they are not tightly integrated with the algorithmic description, which makes it difficult to follow how the components interact.

- The authors state that NES is superior to gradient-based adaptation, but the paper does not provide a deeper narrative or analytical justification beyond empirical observation. There is no analysis of failure modes or why gradient-free adaptation is particularly effective in this setting.

- The experimental section includes many numerical comparisons, but offers very limited interpretation of the observed behavior. The focus is on performance differences rather than understanding the mechanism behind them.

- The authors claim the approach is architecture- and domain-agnostic. The experiments do demonstrate architecture-agnostic behavior to a certain degree, but domain-agnostic generality is not evidenced, as all evaluations are restricted to JSS/FJSS scheduling problems.

**Questions:**

- Could you provide evidence or analysis showing why gradient-based adaptation fails and identify cases where gradient-based adaptation performs similarly or better? Additionally, what properties of the problem or model determine when NES has an advantage over gradient-based optimization?

- Could you provide a more detailed efficiency analysis such as wall-clock runtime breakdown, GPU utilization, and the contributions of population size and sampling to compute cost? Additionally, is the training overhead representative for inference as well, or does inference scale differently?

- Since the computational cost of NES scales roughly linearly with population size, could you provide an ablation study showing how different values of the population size affect solution quality, runtime, and adaptation stability?

- Do you have an idea, which characteristics a problem must have for your method to be effective?

- The empirical section shows strong quantitative improvements. However, you do not provide qualitative insights into how the adapted decisions change. Could you include a qualitative or visual analysis showing how adaptation alters decisions and why it leads to better performance?

---

> ### Author Response · Authors · 2025-11-23
>
> We greatly appreciate your support and positive evaluation of our work. Below we provide our responses to your concerns.
>
> **W1-Improved modularity and clarity:** We sincerely appreciate the reviewer’s valuable feedback. We acknowledge that the paper introduces multiple complex components simultaneously, and we understand the concern regarding the cognitive load caused by the heavy notation and lack of clear modularization. We will carefully revise the paper to better modularize the explanations and ensure that the conceptual flow is more clearly established before introducing detailed notation. We will also update the manuscript to ensure that the figures are more closely tied to the corresponding explanations. We are grateful for the reviewer’s valuable suggestions and believe these changes will significantly improve the clarity and accessibility of the paper.
>
>
> **W2&W3-Deeper analysis of why NES works:** Thank you for this important comment. We believe the advantage of NES over gradient-based method in our case is due to the following reasons.
>
> **First**, for the RL agent, solving FJSP and other combinatorial optimization problem is a task with *sparse reward*, since the reward signal is only generated at the end of each episode when a complete solution is constructed and the objective value is obtained. Gradient-based methods are known to struggle under sparse reward settings. To alleviate this issue, existing gradient-based scheduling methods such as (Zhang et al., 2020; Song et al., 2023; Wang et al., 2023) shape the reward to be denser by designing it as the makespan increment between two consecutive steps. However, the reward is still relatively sparse because not all actions lead to a makespan increment. We perform a quick test by running the DANIEL model on 100 instances from the 10×5 SD2 distribution, and found that on average only 15% timesteps in one episode have non-zero rewards. Therefore, the sparse reward issue largely remains. In contrast, NES methods like ours directly work with sparse reward without the need of shaping, since the quality of the policy is simply evaluated based on complete solutions.
>
> To support the above analysis, we replace the reward of FOMAML to the sparse reward used in our model (i.e., a non-zero reward is obtained at the end of each episode). Results on the 10×5 SD2 dataset (Table R1) show that under this setting, the performance of gradient-based meta-learning further deteriorates, which validates our analysis.
>
> **Table R1.** Impact of Reward Sparsity on Gradient-Based Fine-Tuning
> ||Ours|FOMAML|FOMAML (sparse reward)|
> |:-:|:-:|:-:|:-:|
> |Greedy|369.2|394.4|400.8|
> |Sampling|347.7|356.0|361.2|
>
> **Second**, the fine-tuning process on an instance can be viewed as a search procedure for solving FJSP. Like other COPs, the solution space of FJSP is highly dicrete, non-smooth and non-convex. Gradient-based method would encounter substantial difficulties in this case, since the gradient could be highly unstable and the search could easily be trapped in local optimum. In contrast, NES methods like ours could provide more robust gradient estimation for combinatorial search space, and have stronger global search ability hence is less prone to local optimum.
>
> To demonstrate this, we present the 10-step fine-tuning process of two representative instances with 13.6% and 25.8% non-zero reward in Tables R2 and R3, respectively. We can observe that on both instances, FOMAML easily gets trapped in local optimum, and the one with sparse reward performs even worse. EAS performs the best among gradient-based methods, since it directly use the objective function to update parameters and is less affected by the sparse reward issue, but it is still prone to local optimum. In contrast, the makespan of our method drops stably and is eventually lower than all gradient-based methods, showing a stronger search ability.
>
> **Table R2.** Instance-wies fine-tuning process (13.6% Non-zero Reward)
> ||step1|step2|step3|step4|step5|step6|step7|step8|step9|step10|
> |-|-|-|-|-|-|-|-|-|-|-|
> |Ours|460|452|466|485|452|437|439|421|421|421|
> |EAS|464|510|510|433|479|479|479|433|433|433|
> |FOMAML|463|463|463|463|479|479|479|479|479|479|
> |FOMAML (sparse reward)|463|463|463|480|480|480|480|491|491|480|
>
> **Table R3.** Instance-wies fine-tuning process (25.8% Non-zero Reward)
> ||step1|step2|step3|step4|step5|step6|step7|step8|step9|step10|
> |-|-|-|-|-|-|-|-|-|-|-|
> |Ours|454|422|388|394|386|373|368|373|366|368|
> |EAS|436|436|396|396|396|377|377|377|377|377|
> |FOMAML|439|439|439|439|402|402|402|402|394|394|
> |FOMAML (sparse reward)|439|439|439|439|439|436|436|439|439|439|

---

> > ### Author Response · Authors · 2025-11-23
> >
> > **W4-Broader applicability:** Thanks for this important comment. Methodologically, our derivative-free method is agnostic to model architecture or learning paradigm and only requires querying the objective function as a black box, hence is can potentially be applied to other combinatorial optimisation problems with well‑defined objective functions.
> >
> > We successfully demonstrate the versatility of our method by applying it to two fundamental problems JSP (appendix) and FJSP, on two SOTA learning models with different architectures and training schemes.
> >
> > We fully agree that demonstrating broader applicability to other domains would further enhance the impact. We are actively exploring such extensions and believe that our framework can transfer to those settings with appropriate problem-specific modifications.
> >
> > **Q1-Deeper analysis of why NES works:** Please see the above response to **W2&W3**.
> >
> >
> > **Q2-Efficiency analysis:** We thank the reviewer for this comment. We have added a detailed efficiency analysis on the SD2 distributions, with Table R4 providing a breakdown of the wall-clock time to better understand the computational costs of each stage. Tables R5 and R6 show the GPU utilization during training for instances of different sizes, as well as the GPU utilization for different population sizes in the 10x5 instance. As the size or population increases, the GPU utilization accordingly increases. Our inference phase is actually the same as the inner-loop process in the training phase, so the computational cost scales similarly.
> >
> > **Table R4.** Breakdown of wall-clock time for training phases
> > ||10×5|20×5|15×10|20×10|
> > |-|-|-|-|-|
> > |Environment Initialization|34.1%|32.7%|24.3%|19.0%|
> > |Environment Update|18.9%|18.5%|14.1%|12.8%|
> > |Forward Propagation|44.3%|47.5%|60.1%|66.4%|
> > |Other|2.7%|1.3%|1.5%|1.8%|
> >
> >  **Table R5.** GPU utilization during training for different instance sizes ($\mu=100$)
> > ||10×5|20×5|15×10|20×10|
> > |-|-|-|-|-|
> > |GPU Utilization|49%|74%|85%|88%|
> >
> > **Table R6.** GPU utilization for different population sizes on the 10x5 distribution
> > ||$\mu=20$|$\mu=50$|$\mu=100$|$\mu=300$|
> > |-|-|-|-|-|
> > |GPU Utilization|24%|33%|49%|84%|
> >
> >
> > **Q3-Ablation study on population size:** Thank you for this comment. To further investigate the impact of population size, we performed additional ablation experiments on the 10×5 SD2 set (the same set used for hyperparameter tuning). As illustrated in Table R7, a larger population improves performance but also leads to a significant increase in training time. To balance solution quality and training cost, we set the population size to $\mu=100$.
> >
> > **Table R7.** Impact of population size $\mu$
> > ||$\mu=20$|$\mu=50$|$\mu=100$|$\mu=300$|
> > |-|-|-|-|-|
> > |Performance|377.5|374.2|369.2|365.5|
> > |Training time|1.7h|1.8h|2.1h|3.3h|
> >
> >
> > **Q4-Characteristics of problems for our method to be effectiveness:** Thank you for this valuable comment. The effectiveness of our method, which combines derivative-free optimization with meta-learning for instance fine-tuning, is particularly evident in problem domains where gradient-based optimization faces limitations. More specifically:
> >
> > - Our method could be effective when the underlying RL problem suffers from sparse reward issue, which could impede the performance of gradient-based methods. In addition, our method is also suitable for COPs where the landscape of objective function is highly irregular, since gradient-based methods are prone to local optimum due to the weak global search ability.
> >
> > - Our method could be effective on large-scale problems, which is challenging for gradient-based methods because: 1) gradients need to be stored, which significant affets its scalablity as shown in Table 8 in our experiments; and 2) the huge search space of each instance cannot be effectively explored by the local search mechanism of gradient-based methods during fine-tuning.
> >
> > - Our method could be effective in dynamic environments where tasks may change over time, or the problem setup may vary. Our meta-learning approach allows the model to generalize better across diverse task distributions, while derivative-free optimization ensures stability in the absence of reliable gradient information.
> >
> >
> > **Q5-Instance adaptation visualization:** Thanks for the valuable comment. We visualize the decision changes before and after instance adaptation using Gantt charts. We selected a 10x5 instance and compared the Gantt charts before and after fine-tuning, shown as Figures 4 and 5 in the revised Appendix F. In Figure 4 (before fine-tuning), the initial model fails to achieve optimal performance for the specific instance, with significant machine idle times and relatively large discrepancies in completion times across different machines, leading to a larger makespan. After fine-tuning (Figure 5), the schedule is clearly more compact, effectively reducing idle times and aligning completion times across machines, resulting in a lower makespan.

---

> ### Comment · Reviewer_3mss · 2025-11-27
>
> Thank you very much for the structured rebuttal, as well as the revisions to the manuscript. Several clarifications are helpful for me and in my opinion improve the paper.
>
> Regarding W1: I appreciate the announced restructuring, but in the current revision most of the changes appear concentrated in Section 3. The conceptual narrative in Section 4, where the interaction of the loop structures and the meta-objective should become most transparent, remains largely unchanged. As a result, some of the original concerns about cognitive load and conceptual flow persist.
>
> Regarding W2 and W3: I acknowledge the additional empirical evidence and references to the recognised challenges of gradient-based methods in sparse reward and discrete combinatorial settings. Nevertheless, the contribution here seems largely confirmatory; the arguments rely on well-established phenomena and the paper does not yet offer any deeper analytical insights or comparative evaluations beyond the immediate set of baselines. Reviewer kVaC raised similar concerns, and while the empirical rationale is clearer, the conceptual novelty remains limited.
>
> Regarding W4: The claim of domain-agnostic applicability is interesting, but the rebuttal does not yet provide tangible preliminary evidence from other domains. Offering even small-scale early results or a concrete outline of observed transferability would help substantiate this claim.
>
> Regarding Q3: You provide helpful tables on population size, but it remains unclear how training time decomposes across components and how the trade-off was formally defined. It would be valuable to understand whether larger populations were explored and how quickly diminishing returns appear.
>
> Regarding Q4 and Q5: Thank you for the additional clarifications.
>
> Finally, I share Reviewer kVaC’s remaining concerns about the lack of a compelling argument for conceptual novelty, and the fact that the OpenAI-ES update rule is followed. Thus, while the approach performs well in practice, its theoretical basis could be improved. The paper makes a meaningful empirical contribution, and I welcome the additional experiments and clarifications that have convinced me more towards acceptance of the paper. However, the revision does not fully address the concerns about conceptual novelty, theoretical grounding and improved clarity. My evaluation therefore remains unchanged.

---

> > ### Author Response · Authors · 2025-12-02
> >
> > We greatly appreciate reviewer's thorough evaluation and are pleased that our responses have  convinced you more towards acceptance. Below, we provide additional clarifications regarding the remaining issues raised.
> >
> >
> > **Improved modularity and clarity:** We have thoroughly revised both Section 3 and 4 to make our method more accessible, especially the relationship between the gradients and the meta-objective.
> >
> >
> > **Regarding novelty:** We agree that the challenges faced by gradient-based methods in sparse-reward and combinatorial settings are well established. Our goal is not to introduce new principles about these difficulties, but to demonstrate that they are the key bottlenecks preventing prior meta-learning approaches from functioning effectively on complex scheduling problems.
> >
> > The main contribution of our work is therefore practical and methodological rather than conceptual: we show that a derivative-free meta-learning pipeline can be made effective at scale through a combination of 1) ES-based instance adaptation that consistently outperforms gradient-based fine-tuning, 2) a lightweight first-order meta-update compatible with derivative-free inner loops, 3) variance-reduced multi-sample estimators, and 4) population-level GPU parallelization.
> >
> > Together, these components allow our method to achieve stable, strong performance on substantially larger FJSP instances (50×20 and 100×20) than those tackled in prior learning-based scheduling literature. Additionally, for comparative evaluations beyond the immediate set of baselines, please refer to our response regarding **Domain-agnostic applicability and preliminary evidence of transferability**.
> >
> >
> > **Domain-agnostic applicability and preliminary evidence of transferability:** We sincerely appreciate the reviewer’s valuable feedback. To demonstrate the domain-agnostic applicability of the framework, we further apply our derivative-free meta-learning framework to TSP and CVRP, two well-known routing problems, with 20 and 100 nodes. Specifically, we implement both FOMAML and our derivative-free meta-learning using the seminal POMO (Kwon et al., 2020) as the backbone model. Both FOMAML and our method are initialized using the pre-trained POMO model, so as to benefit from its strong performance and save experiment time. Both methods are trained for the same 50 epochs, followed by 10-step fine-tuning on the same 100 test instances under the single-trajectory mode. Results of these preliminary evaluations are listed in Table R1. It can be observed that our method consistently outperforms gradient-based approaches, showing the domain-agnostic applicability of our framework in COPs beyond scheduling.
> >
> > **Table R1.** Results for TSP and CVRP
> > ||TSP20|TSP100|CVRP20|CVRP100|
> > |-|-|-|-|-|
> > |POMO|3.81|7.74|6.37|16.03|
> > |FOMAML|3.81|7.71|6.23|15.86|
> > |Ours|3.80|7.69|6.18|15.85|
> >
> > **Training Time Decomposition and Population Marginal Benefits:** We sincerely appreciate the reviewer's valuable feedback. To further investigate the impact of population size, we explored larger population sizes on the 10x5 SD2 set. As shown in Table R2, increasing the population size improves performance, at the cost of increasing training time. However, we observe that when the population reaches 500, further increases in population size do not yield significant performance gains. Therefore, we use $\mu=100$ as a practical trade-off between performance and efficiency.
> >
> > **Table R2.** Impact of population size $\mu$
> > ||$\mu=20$|$\mu=50$|$\mu=100$|$\mu=300$|$\mu=500$|$\mu=1000$|
> > |-|-|-|-|-|-|-|
> > |Performance|377.5|374.2|369.2|365.5|364.8|364.3|
> > |Training time|1.7h|1.8h|2.1h|6.2h|11.0h|21.4h|

---

### Official Review · Reviewer_3BUB · 2025-11-01

**Soundness:** 3
**Presentation:** 3
**Contribution:** 3
**Rating:** 6
**Confidence:** 4

**Summary:**

This paper proposes a derivative-free meta-learning framework for solving NP-hard scheduling problems (JSP and FJSP). The approach aims to address two main limitations of existing learning-based scheduling methods: (1) they optimize average performance rather than instance-specific quality, and (2) existing test-time adaptation methods like Active Search (AS) and Efficient Active Search (EAS) use gradient-based optimization and don't share knowledge across instances.

The proposed approach builds on MAML (an efficient DRL-based appraoch) and on Natural Evolution Strategies (NES) for derivative-free optimization

**Strengths:**

1) Novelty: while neither component (MAML and NES) is novel in itself, they are used in a way that is novel for the domain.

2) Comprehensive evaluation: the evaluation covers multiple environment sizes and consists of several public benchmarks.

3) Empirical improvement: the appraoch achieves consistent improvement across multiple experiments.

**Weaknesses:**

1) Lack of statistical significance tests: while the appraoch shows improvement in most cases, in some cases the improvement is small. The standard deviation is not presented (except for one table in the appendix), so it is impossible to determine whether the results are significant. Statistical significance tests would strengthen the authors' claims regarding the superior performance of their approach.

2) Training cost no sufficiently emphasized and discussed: while parallelization is a plus, the training cost is significant. In some cases, training can take more than twice the time of competitors. Moreover, the gap (i.e. the difference in percentages between the proposed approach and FOMAML) seems to grow as the problem grows more complex. This raises significant questions regarding the feasibility of the approach for large-scale problems.

3) No analysis of hyperparameters or sensitivity analysis: No analysis is provided on sensitivity to key parameters like population size (μ=100), noise standard deviation (σ=0.2), or number of inner-loop steps (K=3). How robust is the method to these choices?

**Questions:**

I invite the authors to address the weaknesses mentioned above.

---

> ### Author Response · Authors · 2025-11-23
>
> Thank you so much for the valuable comments. We address your concerns in the following responses.
>
> **W1-Lack of statistical significance testing:** Thank you for the valuable comment. We would like to gently point out that the performance of our algorithm, along with the corresponding standard deviations, is presented in Tables 11, 12, and 13 in the appendix. These tables provide the results for different datasets and configurations, highlighting the average performance improvements and their variability across multiple runs. Additionally, we have conducted further experiments to assess the statistical significance of the improvements observed with our approach. Specifically, we performed paired t-tests to compare the results of our method against the baseline on the the FJSP and JSP datasets under various scenarios.
>
> The results of these significance tests are summarized in Tables R1, R2, and R3. For each case, we report the t-statistic and p-value to provide a clear indication of whether the observed improvements are statistically significant. Our results show that the advantage of our method is statistically significant (p-values below the 0.05 threshold) on most cases, except on the MK dataset in Table R2 where the p-value is slightly higher. This is because the MK dataset contains only 10 instances and the baseline AS/EAS are already close to optimum. We have updated Table 2 in the main paper, where AS and EAS are also marked as bold on the MK dataset.
>
> **Table R1.** Statistical Significance test on FJSP Synthetic Test Sets: t-statistic (p-value)
> ||||10×5|20×5|15×10|20×10|30×10|40×10|
> |-|-|-|-|-|-|-|-|-|
> |SD1|Greedy|AS|-9.3(3e-8)|-7.3(1e-10)|-6.1(3e-8)|-2.9(4e-5)|-7.1(3e-7)|-5.2(3e-8)|
> |||EAS|-3.0(2e-3)|-8.2(2e-10)|-5.8(5e-8)|-3.4(2e-4)|-7.4(2e-9)|-5.6(6e-6)|
> ||Sampling|AS|-8.9(2e-8)|-10.7(4e-8)|-7.1(3e-8)|-11.5(3e-10)|-9.7(2e-10)|-7.8(1e-8)|
> |||EAS|-6.7(8e-10)|-10.4(2e-10)|-6.4(4e-10)|-10.4(7e-10)|-8.6(4e-9)|-5.4(6e-8)|
> |SD2|Greedy|AS|-10.6(3e-8)|-11.2(2e-10)|-9.0(1e-4)|-6.8(9e-7)|-10.2(5e-6)|-3.9(1e-2)|
> |||EAS|-5.6(2e-7)|-9.8(2e-6)|-6.3(5e-9)|-4.6(9e-6)|-4.2(5e-5)|-1.8(3e-2)|
> ||Sampling|AS|-10.2(2e-7)|-18.9(1e-8)|-2.5(1e-2)|-15.6(2e-10)|-15.3(9e-10)|-18.7(1e-10)|
> |||EAS|-8.1(1e-7)|-15.6(1e-8)|-1.2(3e-2)|-12.1(2e-10)|-12.3(7e-10)|-15.7(3e-9)|
>
> **Table R2.** Statistical Significance test on FJSP Public Benchmark Datasets: t-statistic (p-value) (results with * means not significant)
> ||||mk|la(rdata)|la(edata)|la(vdata)|
> |-|-|-|-|-|-|-|
> |10x5|Greedy|AS|-0.4(3e-1*)|-10.2(2e-10)|-11.4(1e-10)|-5.3(3e-7)|
> |||EAS|-0.3(5e-1*)|-7.5(6e-7)|-8.3(3e-10)|-2.7(1e-2)|
> ||Sampling|AS|-3.2(1e-2)|-6.1(3e-8)|-11.9(2e-10)|-4.1(2e-7)|
> |||EAS|-2.2(2e-2)|-5.8(1e-7)|-10.2(3e-9)|-3.0(4e-3)|
> |15x10|Greedy|AS|-0.5(5e-1*)|-7.3(4e-7)|-10.5(9e-10)|-4.9(2e-5)|
> |||EAS|-0.5(6e-1*)|-6.3(2e-6)|-8.3(6e-9)|-3.8(3e-4)|
> ||Sampling|AS|-3.8(1e-2)|-7.9(7e-8)|-10.4(1e-9)|-2.3(4e-5)|
> |||EAS|-1.9(3e-2)|-6.9(4e-7)|-8.9(4e-10)|-1.8(2e-3)|
>
> **Table R3.** Statistical Significance test on JSP Taillard's benchmark Datasets: t-statistic (p-value)
> |||15×15|20×15|20×20|30×15|30×20|
> |-|-|-|-|-|-|-|
> |Greedy|AS|-7.5(9e-7)|-6.3(3e-3)|-10.8(2e-10)|-7.9(3e-8)|-5.8(1e-3)|
> ||EAS|-6.3(4e-6)|-3.2(2e-3)|-9.3(6e-9)|-5.3(1e-7)|-3.5(2e-3)|
> |Sampling|AS|-3.9(3e-4)|-5.1(2e-5)|-1.9(3e-5)|-5.0(5e-5)|-5.3(6e-6)|
> ||EAS|-2.6(1e-3)|-5.3(4e-5)|-2.6(1e-5)|-4.5(3e-4)|-2.8(2e-5)|

---

> > ### Author Response · Authors · 2025-11-23
> >
> > **W2-Training cost and scalability concerns:** Thank you for the important comment. We would like to clarify your concern in the following aspects.
> >
> > **First**, the low training cost of FOMAML is somewhat misleading. Combining the results in Table 5 and Table 1, we can see that the majority of FOMAML's performance is inferior to that of EAS and AS which do not employ meta-learning. In other words, its fast training is actually of no significant meaning here since it is counterproductive to perform gradient-based meta-learning.
> >
> > **Second**, while our method in the paper is trained for 200 epochs, its convergence is fast, achieving satisfactory results early in training. Here we list the performance of our method after 100 epochs on SD2 in Table R4, which shows that our model trained for 100 epochs performs nearly identically to the one trained for 200 epochs, while the training time is only half as in Table R5. So in practice, early stopping is possible to obtain good performance.
> >
> > **Table R4.** Comparison of solution quality
> > ||10×5|20×5|15×10|20×10|30×10|40×10|
> > |-|-|-|-|-|-|-|
> > |FOMAML|394.4|646.6|557.1|583.8|757.6|949.3|
> > |Ours(100 Epochs)|369.2|625.1|531.8|561.7|732.8|920.5|
> > |Ours(200 Epochs)|369.2|624.8|531.4|559.9|732.6|920.1|
> >
> > **Table R5.** Comparison of training time
> > ||10×5|20×5|15×10|20×10|
> > |-|-|-|-|-|
> > |FOMAML|1.5h|2.9h|3.7h|5.5h|
> > |Ours(100 Epochs)|1h|2.3h|3.6h|5.5h|
> > |Ours(200 Epochs)|2.1h|4.5h|7.2h|11.2h|
> >
> > **Finally**, at inference time, our method can scale to very large-scale problems, as shown in Table 8 of the appendix which listed results on very large FJSP instances (sizes 50×20 and 100×20) using the SD2 distribution. Results demonstrate that our meta-model exhibits strong generalization performance on these large-scale instances. In both greedy and sampling modes, its zero-shot performance already surpasses the 1-hour results of Ortools, and fine-tuning further boosts its performance, outperforming all baseline methods comprehensively.
> >
> > **W3-Lack of hyperparameter sensitivity analysis:** Thank you for the valuable comment. Here we examine the impact of four important hyperparameters in NES: the population size $\mu$, the standard deviation $\sigma$ of the search distribution, the number of inner-loop steps $K$ and the update step sizes $\alpha$ and $\beta$. The experiments are conducted on the 10×5 SD2 set (the one we used for hyperparameter tuning). As shown in Table R6, a larger population leads to better performance but also increases training time. We set $\mu=100$ to achieve a reasonable tradeoff between solution quality and training cost. Table R7, R8, and R9 show that the impacts of $\sigma$, $K$, $\alpha$, and $\beta$ are relatively small, with $\sigma = 0.2$, $K = 3$, and $\alpha = \beta = 0.05$ yielding the best performance.
> >
> > **Table R6.** Impact of population size $\mu$
> > ||$\mu=20$|$\mu=50$|$\mu=100$|$\mu=300$|
> > |-|-|-|-|-|
> > |Performance|377.5|374.2|369.2|365.5|
> > |Training time|1.7h|1.8h|2.1h|3.3h|
> >
> > **Table R7.** Impact of standard deviation $\sigma$
> > ||$\sigma=0.05$|$\sigma=0.1$|$\sigma=0.15$|$\sigma=0.2$|$\sigma=0.25$|$\sigma=0.3$|
> > |-|-|-|-|-|-|-|
> > |Performance|375.6|372.3|372.6|369.2|374.2|375.4|
> > |Training time|2.1h|2.1h|2.1h|2.1h|2.1h|2.1h|
> >
> > **Table R8.** Impact of inner-loop steps $K$
> > ||$K=1$|$K=2$|$K=3$|$K=4$|$K=5$|
> > |-|-|-|-|-|-|
> > |Performance|372.9|369.7|369.2|370.6|370.2|
> > |Training time|1.2h|1.7h|2.1h|2.6h|3.1h|
> >
> > **Table R9.** Impact of step sizes $\alpha$ and $\beta$
> > || $\alpha = \beta = 0.03$| $\alpha = \beta = 0.04$| $\alpha = \beta = 0.05$| $\alpha = \beta = 0.06$| $\alpha = \beta = 0.07$|
> > |:-:|:-:|:-:|:-:|:-:|:-:|
> > |Performance|374.3|370.5|369.2|372.5|370.9|
> > |Training time|2.1h|2.1h|2.1h|2.1h|2.1h|

---

### Meta-Review · Area_Chair_4fms · 2026-01-07

**Summary:**

Existing learning-based schedulers primarily optimize average performance across training instances, often compromising instance-specific solution quality. Meanwhile, test-time adaptation methods typically rely on gradient-based optimization, which is poorly suited to combinatorial search spaces. Motivated by these limitations, this paper proposes a derivative-free, instance-wise meta-learning framework for NP-hard scheduling problems, with a particular focus on job-shop scheduling (JSP) and flexible job-shop scheduling (FJSP).

The reviewers generally agreed on the following strengths:
- S1. Although the individual components (MAML and NES) are not novel, their combination and adaptation to scheduling problems are novel and well motivated for this domain.
- S2. The experimental study is extensive, covering multiple environment sizes and a range of standard benchmarks.
- S3. The proposed approach demonstrates consistent performance improvements across most experimental settings.

The main weaknesses identified by the reviewers include:
- W1. Additional experiments are needed to assess efficiency, hyperparameter sensitivity, and statistical significance.
- W2. The paper occasionally uses imprecise terminology, and its relationship to prior work is not always clearly articulated.
- W3. The method lacks a clear theoretical justification.

During the rebuttal phase, the authors largely addressed W1 and partially addressed W2 and W3.

Despite the remaining limitations, the strengths of the paper outweigh its weaknesses, and acceptance is recommended.

**Reviewer Concerns:**

W1 has been largely addressed. For W2 and W3, the authors made meaningful improvements, such as clarifying the connections to and applicability of existing theoretical results; however, Reviewer kVaC explicitly noted that these concerns were not fully resolved.

**Reviewer Scores:**

Given the extensive rebuttal, there maybe a slight positive adjustment.

---

### Decision · Program_Chairs · 2026-01-26

Accept (Poster)